# Multi-Machine Learning Ensemble Regionalization of Hydrological Parameters for Enhancing Flood Prediction in Ungauged Mountainous Catchments

Kai Li, Linmao Guo, Genxu Wang*, Jihui Gao*, Xiangyang Sun, Peng Huang, Jinlong Li, Jiapei Ma, Xinyu Zhang

*State Key Laboratory of Hydraulics and Mountain River Engineering, College of Water Resources and Hydropower, Sichuan University, Chengdu, 610000, China*

*Corresponding author: Genxu Wang (wanggx@scu.edu.cn) and Jihui Gao (jgao@scu.edu.cn).

## Abstract:

Machine learning-based parameter regionalization is an important method for flood prediction in ungauged mountainous catchments. However, single machine learning parameter regionalization often exhibits limitations in prediction accuracy and robustness. Therefore, this study proposes a multi-machine learning ensemble regionalization method that integrates Gradient Boosting Machine (GBM), K-Nearest Neighbors (KNN), and Extremely Randomized Trees (ERT) methods (GBM-KNN-ERT) to regionalize the sensitive parameters of the Topography-Based Subsurface Storm Flow (Top-SSF) model. Validated across 80 mountainous catchments in southwestern China, the GBM-KNN-ERT method demonstrates superior performance with 90% of ungauged catchments achieving the Nash-Sutcliffe Efficiency (NSE) above 0.9, representing a 67.44% improvement over the best single machine learning parameter regionalization. Notably, the GBM-KNN-ERT method shows improved robustness to climate change and changes in the number of donor catchments compared to other regionalization methods. An optimal balance between accuracy and computational efficiency was achieved using 20-40 high quality donor catchments

(NSE greater than 0.85). This study provides systematic evidence that multi-machine
learning ensemble can effectively address regionalization challenges in ungauged
mountainous regions, offering a reliable tool for water resource management and flood
disaster mitigation.
**Keywords**: Flood prediction; Regionalization; Ungauged mountainous catchments;
Top-SSF model;
**Highlights:**
1. Proposes a novel multi-machine learning ensemble regionalization method
2. The GBM-KNN-ERT method increases the percentage of catchments with high-
accuracy flood predictions (NSE >0.9) to 90%, which is a 67.44% improvement
over the best single machine learning method.
3. The GBM-KNN-ERT method exhibits greater stability under climate change.

# 1. Introduction

Floods in mountainous catchments, encompassing both flash floods and general larger-scale flood events which can be derived from mountainous upland catchments, pose a significant threat to human safety and property, particularly in regions lacking sufficient observational data (Luo et al., 2015; Zhai et al., 2018). While hydrological models like the Topography-Based Subsurface Storm Flow (Top-SSF) model (Li et al., 2024) offer promising simulation capabilities, their application in ungauged catchments is severely limited by the absence of calibration data (Choi et al., 2023; Liu et al., 2018). Effective parameter regionalization methods are therefore essential for transferring hydrological knowledge from gauged to ungauged regions, enabling reliable flood prediction in ungauged mountainous catchment (Garambois et al., 2015; Ragettli et al., 2017; Xu et al., 2018).

Parameter regionalization is a crucial method for flood prediction in ungauged catchments (Arsenault et al., 2023; Guo et al., 2021; Kratzert et al., 2019; Zhang et al., 2020). Compared to purely data-driven methods, parameter regionalization offers enhanced physical interpretability (Nearing et al., 2024; Tang et al., 2023; Zhang et al., 2024). Existing parameter regionalization methods can be broadly classified into three categories: similarity-based, hydrological signatures-based, and regression-based (Arsenault et al., 2019; Wu et al., 2023). Similarity-based methods rely on the assumption that catchments with similar characteristics exhibit similar hydrological responses, considering spatial proximity (Arsenault et al., 2019; Pugliese et al., 2018; Yang et al., 2018) and physical similarity (similar climatic and land cover conditions

have similar hydrological characteristics) (Kanishka and Eldho, 2017; Papageorgaki
and Nalbantis, 2016). Hydrological signature-based methods use hydrological
signatures (quantitative metrics that describe statistical or dynamic properties of
streamflow) as an intermediate link, establishing relationships first between model
parameters and signatures, and then between signatures and catchment descriptors to
facilitate parameter transfer (McMillan, 2021; Zhang et al., 2018). Regression-based
methods, which directly link hydrological model parameters to catchment descriptors,
are widely used due to their simplicity and computational efficiency (Guo et al., 2021;
Kratzert et al., 2019; Song et al., 2022; Wu et al., 2023). However, the performance of
regression-based methods is frequently constrained by the inherent nonlinearity in the
relationships between model parameters and catchment descriptors, coupled with the
difficulty in adequately capturing spatial heterogeneity, especially within complex
mountainous terrain (Wu et al., 2023).
Recent advances in machine learning offer potential solutions by capturing
nonlinear patterns in high-dimensional data. Such as Decision Tree (DT), Extremely
Randomized Trees (ERT), Gradient Boosting Machine (GBM), K-Nearest Neighbor
(KNN), Random Forest (RF), and Support Vector Machines (SVM) have shown
promise in parameter regionalization (Golian et al., 2021; Song et al., 2022). However,
existing machine learning-based parameter regionalization studies predominantly focus
on runoff prediction at coarser temporal scales (daily or monthly) (Li et al., 2022; Wu
et al., 2023), leaving a significant gap in high-resolution (hourly or sub-hourly) flood
prediction in ungauged mountainous catchments. Moreover, these studies often rely on
single machine learning methods to estimate all hydrological model parameters (Golian
et al., 2021; Song et al., 2022; Wu et al., 2023). Given that different machine learning
methods operate on distinct principles (Jordan and Mitchell, 2015; Zounemat-Kermani
et al., 2021) and hydrological model parameters represent diverse hydrological
processes (Li et al., 2024), a single machine learning method may not adequately
capture the complexity of model parameter estimation (Golian et al., 2021; Wu et al.,
2023). Therefore, exploring the multi-machine learning ensemble methods is essential
to improve the accuracy of high-resolution flood prediction in ungauged mountainous
catchments.

Southwest China's mountainous regions are particularly vulnerable to frequent

floods, leading to ecosystem degradation through habitat disruption and biodiversity
loss (Gan et al., 2018). The abundance of ungauged catchments in this region poses a
significant challenge to reliable flood prediction. To address this critical issue, we
systematically evaluate the performance of a novel multi-machine learning ensemble
method for regionalizing Top-SSF model parameters across 80 representative
catchments (mean area: 1,586 km²) in Southwest China. By assessing ensemble method
robustness under climate change and with varying donor catchment configurations, this
study aims to significantly enhance flood prediction accuracy in ungauged mountainous
catchments, contributing to improved ecosystem resilience, enhanced human safety,
and more effective water resource management in the face of escalating climatic
pressures.

## 2.  Study area and datasets

### 2.1. Study area

This study investigated 80 mountainous catchments in Southwestern China,
encompassing Sichuan, Yunnan, Guangxi, Guizhou, and Chongqing provinces (Fig. 1).
This region exhibits diverse climatic zones, including subtropical monsoon, plateau
mountain, and tropical monsoon climates. The selected catchments have an average
area of 1,586 km² (ranging from 109 to 6,564km$^2$), with elevations ranging from 63 to
6,284 meters. Mean annual temperature varies from 15 to 20°C, and annual
precipitation ranges from 1,200 to 1,800 mm (Li et al., 2016), with approximately 80%
of the annual precipitation occurring during summer and autumn, contributing to
frequent flooding events (Cheng et al., 2019). These catchments are situated within a
heavily forested region, the second largest in China (Hua et al., 2018), with forest cover
ranging from 3% to 92% (mean: 51%), influencing evapotranspiration and runoff
generation. Dominant soil types, according to the Genetic Soil Classification of China
(Shi et al., 2004), include purple soil (12.20%), yellow soil (11.39%), and red soil
(9.52%), each with distinct hydrological properties.

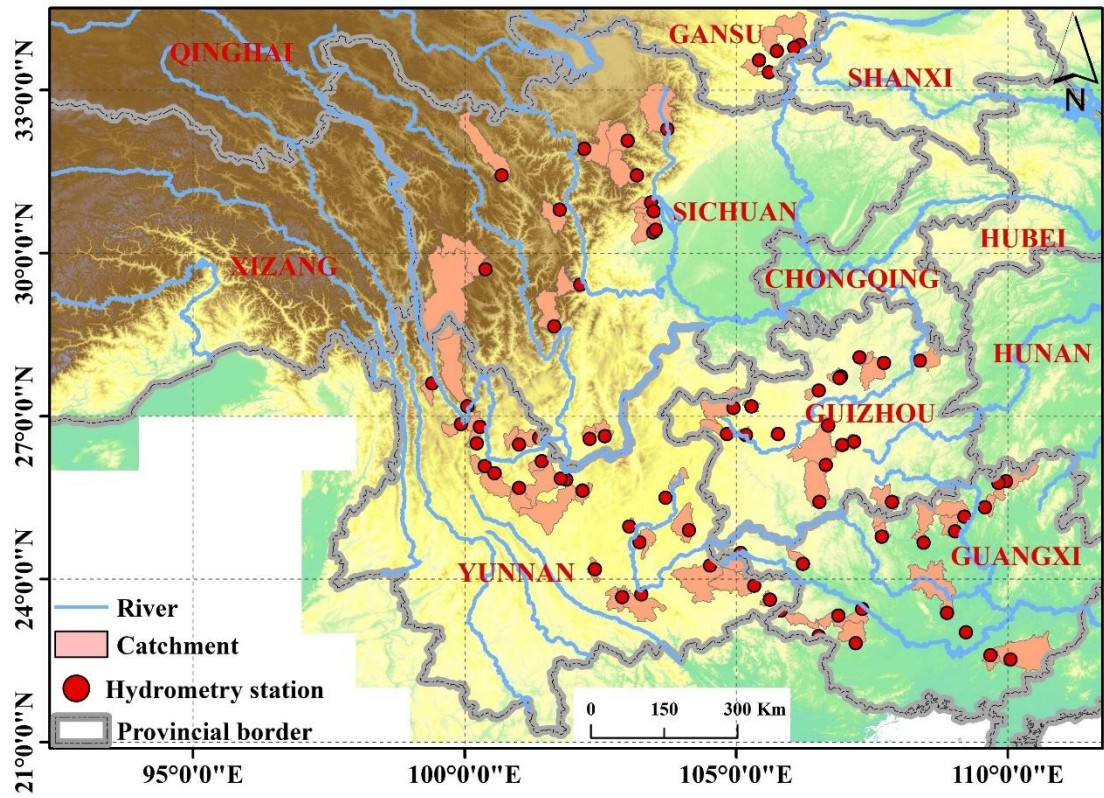


**Fig.1.** Geographical distribution of the 80 gauged catchments used, with locations of hydrometry station (red points) and major rivers indicated.

## 2.2. Datasets

Hourly flow data (2015–2018) for 80 mountainous catchments in China were sourced from the Hydrological Bureau of the Ministry of Water Resources, through China's hydrologic yearbooks, encompassing a spectrum of events from flash floods and general floods which can be derived from mountainous upland catchments. Hourly rainfall data (2015–2018) were obtained from ground meteorological stations across China (http://en.weather.com.cn), providing crucial input for hydrological modelling. Additional meteorological variables, including temperature, wind speed, dewpoint temperature, and surface net solar radiation, were obtained from the ERA5 hourly dataset (1940–present) (Hersbach et al., 2023), ensuring comprehensive atmospheric forcing. Relative humidity was estimated using dewpoint temperature. Historical

(1901–2021) and projected future (SSP585, 2022–2100) temperature and precipitation
data for China, averaged from the EC-Earth3, GFDL-ESM4, and MRI-ESM2-0 models
at 1 km resolution, were obtained from "A Big Earth Data Platform for Three Poles" to
assess the impact of climate change (Ding and Peng, 2020) (http://poles.tpdc.ac.cn).
Topographic data, including a 30-m resolution Digital Elevation Model (DEM), used
for river network and topographic index derivation, were obtained from EARTHDATA
and used for river network delineation and topographic index derivation
(https://search.earthdata.nasa.gov/search). Forest cover data (30-m resolution) were
sourced from the Global Forest Cover and Forest Change Map
(https://www.noda.ac.cn/), providing information on vegetation characteristics. Bulk
density (BD) data were derived from the Soil Database of China for Land Surface
Modelling (Dai et al., 2013). Soil hydraulic parameters, specifically saturated hydraulic
conductivity (Ks_CH) for Clapp and Hornberger functions and the pore-connectivity
parameter (L) for van Genuchten and Mualem functions, were acquired from the China
Dataset of Soil Hydraulic Parameters Using Pedotransfer Functions for Land Surface
Modeling (Shangguan et al., 2013).





**Table 1.** Model forcing data and catchment descriptors information.

| Data type | Name | Unit | Function |
|---|---|---|---|
| | Rainfall | mm | Input for hydrological model |
| | Flood | $m^3/s$ | Used for model calibration (hourly resolution) |
| Hydro-meteorology | Temperature | K | |
| | Surface pressure | Pa | |
| | Dewpoint temperature | K | Input for hydrological model |
| | wind speed | m/s | |
| | Surface net solar radiation | $j/m^2$ | |
| | Relative humidity | % | |
| | 1 km monthly precipitation (1901-2021) | mm | |
| | 1 km monthly temperature (1901-2021) | ℃ | |
| | 1 km monthly temperature (2022-2100, SSP5-8.5, EC-Earth3, GFDL-ESM4, MRI-ESM2-0) | ℃ | Multi-year surface average as catchment descriptors |
| | 1 km monthly precipitation (2022-2100, SSP5-8.5, EC-Earth3, GFDL-ESM4, MRI-ESM2-0) | mm | |
| Soil characteristics | Soil bulk density (BD) | $g/cm^3$ | |
| | Pore-connectivity parameter (L) for the van Genuchten and Mualem functions | - | |
| | Saturated hydraulic conductivity (Ks_CH) of the Clapp and Hornberger Functions | cm d$^{-1}$ | Surface average as catchment descriptors |
| Topography | Forest cover (FC) | % | |
| | DEM | m | |
| | Topographic index | - | |
| | Slope | mm$^{-1}$ | |
| | Catchment area | $km^2$ | |

# 3. Methodology
## 3.1. Hydrological model
Top-SSF is a semi-distributed hydrological model based on the well-established
TOPMODEL framework, which delineates sub-basins based on the topographic index.
It retains the key advantages of TOPMODEL, such as its parsimonious structure,
physical interpretability, and ease of parameter transfer (Beven et al., 2021; Gao et al.,
2018), consists of 15 parameters representing six key hydrological components: canopy
interception, infiltration, evapotranspiration, unsaturated zone moisture transport,
subsurface storm flow, and flow routing (Li et al., 2024). In the Top-SSF model, flood
can be comprised of four components: infiltration-excess overland flow, saturation-
excess overland flow, subsurface storm flow, and groundwater discharge.

Infiltration-excess overland flow occurs when the rainfall intensity exceeds the

infiltration capacity. In this study, infiltration is simulated using the Green-Ampt model.
When surface ponding occurs, the infiltration rate is determined by solving the Green-
Ampt equation iteratively, for which the Newton-Raphson method is employed. The
infiltration rate ($f_{in}$) is given by:
$$f_{in} = -\frac{Ks(CD + F_{satrt})}{Szm(1 - e^{(F_{satrt}/Szm)})} \quad (1)$$
where, $f_{in}$ is the infiltration rate (m/h);$Ks$ is surface hydraulic conductivity (m/h);$CD$
is capillary drive (m); $F_{satrt}$ is the initial cumulative infiltration (m); $Szm$ is the
maximum water storage capacity in the unsaturated zone (m).

Saturation excess overland flow occurs at computational cell $i$ when the

groundwater table depth, $S_i$ is less than or equal to zero (i.e., $S_i \leq 0$, indicating the
water table has reached the surface). It is calculated as:
$$r_{s,i} = max\{Suz_i - max(S_i, 0), 0\} \quad (2)$$
where, $r_{s,i}$ is the depth of saturation excess overland flow generated at cell $i$ (m);
$Suz_i$ is the soil water storage in the unsaturated zone, at cell $i$ (m); $S_i$ is the
groundwater table depth at cell $i$ (m).

The depth of subsurface storm flow generated at computational cell $i$ , $r_{sf,i}$ is

given by:
$$r_{sf,i} = q_{sf0}(1 - S_{sf,i}/S_{fmax}) \quad (3)$$
where, $r_{sf,i}$ is the depth of subsurface storm flow at cell $i$ (m); $q_{sf0}$ is initial
subsurface storm flow (m); $S_{sf,i}$ is the water storage deficit in the subsurface storm
flow zone at cell $i$ (m).

The depth of groundwater discharge is calculated as:

$$r_b = e^{\ln Te - \lambda - \overline{S}_g/Szm} \qquad (4)$$
where, $r_b$ is depth of groundwater discharge (m); $lnTe$ is the log of the areal average
of $T0$ (m$^2$/h); is the catchment average topographic index; $\overline{S}_g$ is the catchment
average groundwater table depth (m). For the complete set of equations for the Top-
SSF model, the reader is referred to the Supplementary Material and Li et al. (2024).
**3.2. Multi-machine learning ensemble method**

To improve flood prediction accuracy in ungauged mountainous catchments, we

proposed a multi-machine learning ensemble method for regionalizing sensitive
parameters of the Top-SSF model. This method leverages the complementary strengths
of multi-machine learning methods to estimate model parameters based on catchment
descriptors (Fig. 2). The characteristics, strengths, and limitations of each machine
learning method are summarized in Table 2. The ensemble method employs a cross-
validation procedure to select the best-performing machine learning method for each
sensitive parameter. These selections are then integrated into a unified regionalization
scheme. By mitigating limitations inherent in single machine learning regionalization,
such as model bias and overfitting, and by capturing complex hydrological processes
in mountainous catchment, this ensemble method aims to achieve more accurate flood
prediction in ungauged catchments.

**Table 2.** Seven machine learning model characteristics, advantages and disadvantages.

| Machine learning | Characteristic | Advantage | Disadvantages |
|---|---|---|---|
| DT | A single decision tree hierarchically partitions the data space using a tree structure, with internal nodes representing features, branches representing decision rules, and leaf nodes representing class labels. | High interpretability; Minimal data preprocessing. | Unstable; Tends to overfit. |
| ERT | Construct multiple decision trees with randomly selected feature values and randomly divided nodes (Geurts et al., 2006). | Low overfitting risk; Computational efficiency; Resilient to noise. | Possibility of increased bias; Limited interpretability. |
| GBM | Construct multiple decision trees. Multiple weak learners are trained iteratively and the loss function is optimised using gradient descent, progressively combined into a robust model through the learning rate (Friedman, 2002). | High accuracy for structured data; Robust to outliers; Minimal data preprocessing. | Limited interpretability; Complex adjustments. |
| KNN | It is a non-parametric, instance-based supervised learning algorithm. It operates by finding the K nearest data points in the training data to a given data point and making predictions based on these (Wani et al., 2017). | Simple and easy to implement. Learning process is quick. | Sensitivity to noisy and scale of data. Accuracy can be heavily impacted by the choice of K. |
| RF | A bagging algorithm proposed by Breiman (2001) that uses ensemble learning. Involves training numerous decision trees and aggregating predictions . | Simple and easy to implement; Low computational cost. | Prone to overfitting in noisy regression tasks. |
| SVM | Identifies hyperplanes in high-dimensional spaces to segregate data. The optimal hyperplane maximizes the margin between it and the nearest data points, termed support vectors (Sain, 1996). | Uses kernel functions to address nonlinear classification issues. | Sensitive to noise |


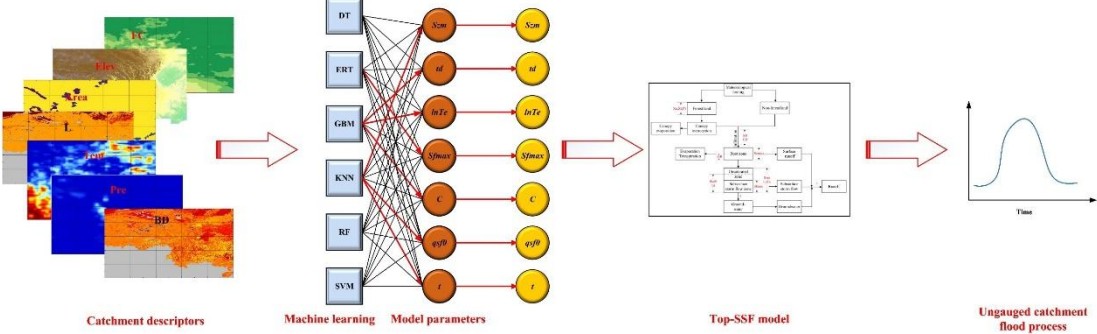

**Fig.2.** Multi-machine learning ensemble method for regionalization in ungauged mountainous catchments. The red line indicates the machine learning method that yielded the optimal parameter estimates.

**3.3. Parameter regionalization process**

The parameter regionalization process comprised four key steps: (1) Top-SSF model calibration and parameter sensitivity analysis; (2) selection of relevant catchment descriptors; (3) establishment of regionalization relationships between sensitive model parameters and catchment descriptors using multi-machine learning ensemble methods; and (4) evaluation of parameter regionalization performance.

**3.3.1. Top-SSF model calibration and parameter sensitivity analysis**

In this study, the Top-SSF model was employed to simulate hydrological processes. The model was driven by continuous hourly meteorological data, including rainfall, temperature, surface pressure, relative humidity, wind speed, and surface net solar radiation. For each catchment, model parameters were calibrated using two hydrologically independent and representative flood events. A third, distinct flood event was then used for model validation. The Nash-Sutcliffe Efficiency (NSE) served as the objective function during calibration, with parameter optimization achieved using the Shuffled Complex Evolution (SCE-UA) algorithm (Duan et al., 1994), known for its global convergence and robustness (Dakhlaoui et al., 2012; Qi et al., 2016). Model

performance was evaluated using the NSE, the relative error of flood peak flow (Qp),
and the absolute error in flood peak occurrence time (Tp), following China's
Specification for Hydrological Information Forecast (GB/T 22482-2008). These
metrics quantify the model's ability to predict flood dynamics, peak flow, and timing.
Following calibration, a sensitivity analysis was conducted to identify and exclude
insensitive model parameters (Lenhart et al., 2002), which were then used for
regionalization. This approach reduces the dimensionality of the regionalization
problem and improves the efficiency of the process.

The sensitivity index ($Si$) of each hydrological model parameter was determined

using the method of Lenhart et al. (2002), which assesses the influence of $\pm 10\%$
changes in parameter values (Eq. 1). Table 3 outlines the sensitivity analysis results for
the model parameters across the 80 mountainous catchments. The $Si$ values are
categorized as follows (Guo et al., 2022): negligible sensitivity ($|Si| < 0.05$),
moderate sensitivity ($0.05 < |Si| < 0.2$), high sensitivity ($0.2 < |Si| < 1.00$), and
extremely high sensitivity ($|Si| \geq 1.00$). Based on the sensitivity analyses, seven
sensitive model parameters were identified: $Szm$, $lnTe$, $Sfmax$, $C$, $qsf0$, $t$ (Table

3).

$$Si = \frac{1}{N}\sum_{t}^{N}\frac{(y_2(t)-y_1(t))/y_0(t)}{2\Delta x/x_0} \qquad (5)$$
where $y_0(t)$ is the flood value of the calibrated parameter $x_0$ at time $t$; $\Delta x$ is the
adjusted parameter difference,$\Delta x/x_0$=10%;$y_1(t)$ is the flood value of the calibrated
parameter $x_0 - \Delta x$ at time $t$;$y_2(t)$ is the flood value of the calibrated parameter
$x_0 + \Delta x$  at time  $t$.

**Table 3.** Top-SSF model main modules and default range of parameters.

| Modular | Parameter | Definition | Unite | Default range | Sensitivity index |
|---|---|---|---|---|---|
| Canopy interception | $Sc$ | Canopy storage capacity | m | 0.00~0.01 | <0.05 |
| | $St$ | Trunk storage capacity | m | 0.00~0.01 | <0.05 |
| | $Pt$ | Proportion of rain diverted into stemflow per cover | % | 0.00~1.00 | <0.05 |
| Evapotranspiration | $Sr0$ | Initial root zone storage deficit | m | 0.00~0.02 | <0.05 |
| | $Srmax$ | Maximum root zone storage deficit | m | 0.00~2 | <0.05 |
| Infiltration | $Ks$ | Surface hydraulic conductivity | m/h | 0~0.01 | <0.05 |
| | $CD$ | Capillary drive (Morel-Seytoux and Khanji, 1974) | m | 0~5 | <0.05 |
| Unsaturated zone | $Suz0$ | Initial baseflow per unit area | m | 0.00~10⁻⁴ | <0.05 |
| | $Szm$ | Soil maximum water storage capacity | m | 0.00~1.00 | **0.19** |
| | $td$ | Unsaturated zone time delay per unit storage deficit | h/m | 0~3 | **1.07** |
| | $lnTe$ | log of the areal average of T0 | m²/h | -2.00~1.00 | **3.4** |
| Subsurface storm flow zone | $Sfmax$ | Maximum subsurface storm flow zone deficit | m | 0.00~0.01 | **0.16** |
| | $C$ | Transfer coefficient | m⁻²/h | 0.00~0.1 | **0.26** |
| | $qsf0$ | Initial subsurface storm flow per unit area | m | 0.00~0.02 | **0.18** |
| Routing | $t$ | Flow routing correction coefficient | - | 0.00~5.0 | **1.21** |

**Note, the bolded values in the sensitivity index indicate sensitive model parameters.**
**3.3.2.    Catchment descriptor selection**

To mitigate the effects of multicollinearity on the accuracy and reliability of the

parameter regionalization methods, catchment descriptors were screened using the
variance inflation factor (VIF) and correlation coefficients. A VIF threshold of less than
10 (VIF < 10) was used to indicate acceptably low multicollinearity (Salmeron et al.,
2018). Initial screening identified strong correlations between several descriptor pairs,
notably L with Ks_CH, and Tem with Elev. Furthermore, the VIF values for Ks_CH
and Slope were found to exceed 10. Consequently, Ks_CH and Slope were removed
from the potential set of descriptors. Following their removal, a re-evaluation of the
VIF for the remaining descriptors was conducted. Although a notable correlation exists
between Tem and elevation (Elev), their VIF values in the reduced set were both below
the threshold of 10. Given the importance of Tem for representing climate impacts and
Elev as a key topographic driver, both were retained to preserve potentially valuable
information. The final set of seven catchment descriptors selected for regionalization
therefore comprised FC, Elev, Area, L, Tem, Pre, and BD. As illustrated in Fig. 3b, the
correlations among these final descriptors and the sensitive model parameters are
generally low (highest at 0.5), suggesting that the relationships are complex and
nonlinear.


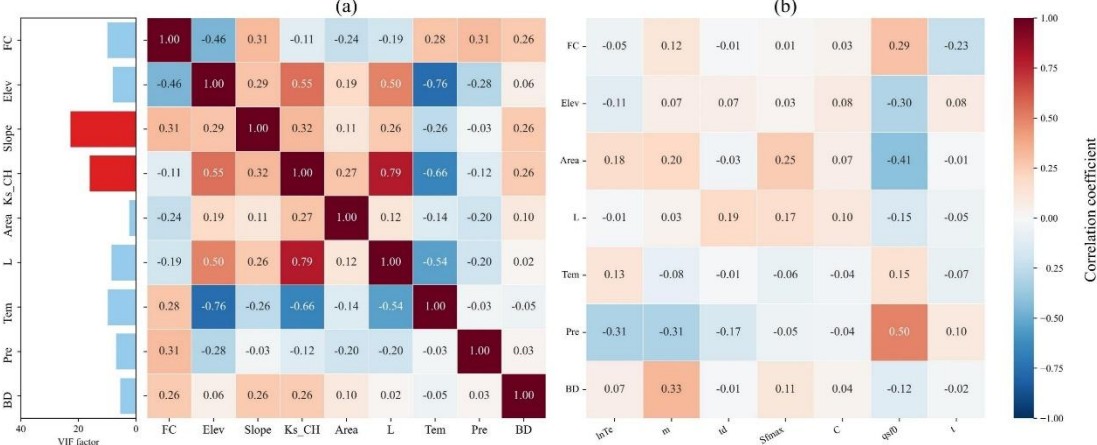

**Fig.3.** Analysis of catchment descriptor relationships: (a) Correlation coefficients and variance inflation factors (VIF) among all descriptors; (b) Correlation coefficients between sensitivity model parameters and descriptors with VIF values below 10.

### 3.3.3. Parameter regionalization

To simulate ungauged catchment conditions, each of the 80 catchments was iteratively treated as an ungauged catchment, with the remaining 79 catchments serving as donor catchments. A parameter regionalization method was then constructed using the catchment descriptors and sensitive model parameters of the donor catchments to predict the seven sensitive model parameters for the ungauged catchment based on its catchment descriptors. These predicted model parameters were then input into the Top-SSF model to enable flood prediction in ungauged catchments. To ensure robust and generalizable results, K-fold cross-validation (K = 10) was implemented. This involved randomly partitioning the 79 donor catchments into K subsets, using one subset as a test set and the remaining K-1 subsets for method training in each iteration (Jung, 2018). This approach maximizes data utilization and minimizes bias associated with specific data partitioning. Hyperparameter tuning for each machine learning method was performed using RandomizedSearchCV (Bergstra and Bengio, 2012), with the objective of minimizing the difference between predicted and observed parameter

values.

### 3.3.4. Evaluated metrics

The performance of the parameter regionalization methods was evaluated by
considering two key aspects. First, the accuracy of the methods in estimating sensitive
model parameters was assessed using three metrics: root mean square error (RMSE),
standard deviation (STD), and the coefficient of determination ($R^2$). The $R^2$ was used
to quantify the agreement between estimated and calibrated parameter sets. Second, to
evaluate the impact of parameter regionalization on flood prediction. The resulting
flood predictions were then evaluated using the NSE, Qp, and Tp metrics.

$$NSE = 1 - \frac{\sum_{j=1}^{M}(Q_{obs}(j) - Q_{sim}(j))^2}{\sum_{j=1}^{M}(Q_{obs}(j) - \overline{Q}_{obs})^2} \quad (6)$$

$$Q_p = \left| \frac{Q_{obs,p} - Q_{sim,p}}{Q_{obs,p}} \times 100\% \right| \quad (7)$$

$$T_p = \left| T_{obs,p} - T_{sim,p} \right| \quad (8)$$

where $Q_{obs}(j)$ is the observed flow rate (m³/s); $Q_{sim}(j)$ is the simulated flow rate
(m³/s); $\overline{Q}_{obs}$ is the mean value of the observed flow rate (m³/s); $Q_{obs,p}$ is the observed
flood peak flow (m³/s); $Q_{sim,p}$ is the simulated flood peak flow (m³/s); $T_{obs,p}$ is the
observed flood peak occurrence time (h); and $T_{sim,p}$ is the simulated flood peak
occurrence time (h).

$$RMSE = \sqrt{\frac{1}{N}\sum_{i=1}^{n}(X_i - Y_i)^2} \quad (9)$$

$$STD = \sqrt{\frac{1}{N-1}\sum_{i=1}^{N}(Y_i - \overline{Y})^2} \quad (10)$$

$$R^2 = \frac{[\sum_{i=1}^{n}(X_i - \overline{X})(Y_i - \overline{Y})]^2}{\sum_{i=1}^{n}(X_i - \overline{X})^2 \sum_{i=1}^{n}(Y_i - \overline{Y})^2} \quad (11)$$

where $X_i$ is the Top-SSF calibration model parameter value; $Y_i$ is the model
parameter estimated value using the parameter regionalization method; $\overline{X}$ and $\overline{Y}$ are
the mean values of $X_i$ and $Y_i$; $N$ is the sample size equal to 80.

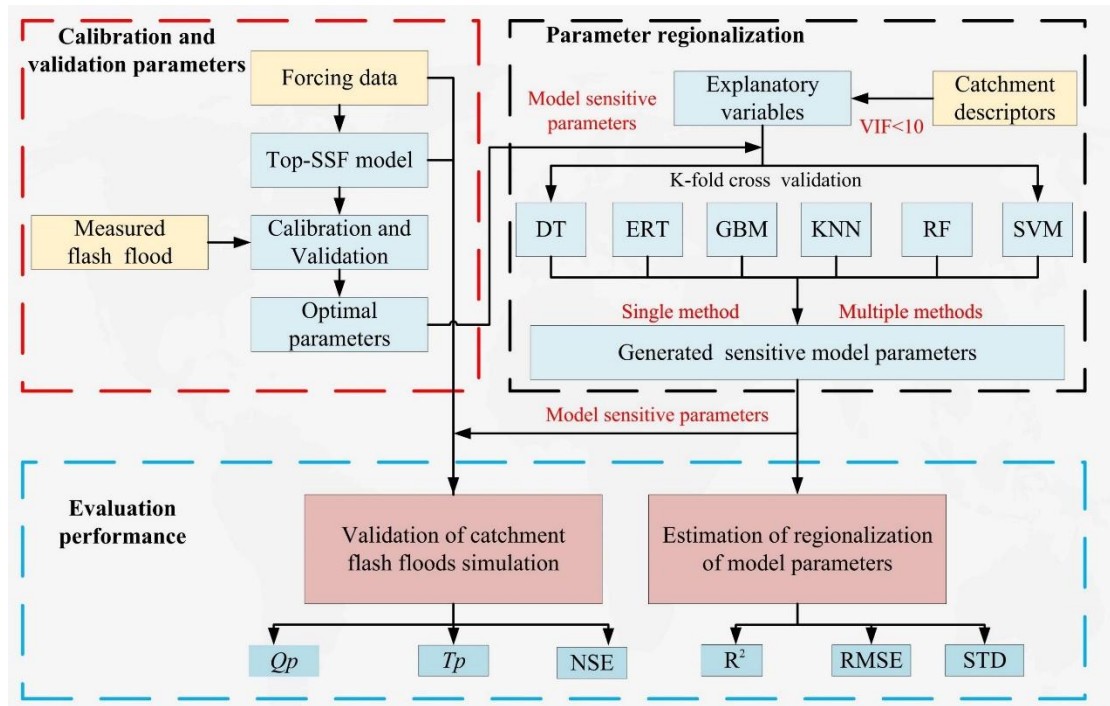


**Fig.4.** Flowchart illustrating the parameter calibration, validation, and regionalization workflow.
Abbreviations: Top-SSF (Topography-Based Subsurface Storm Flow hydrological model),
DT (Decision Tree), ERT (Extremely Randomized Trees), GBM (Gradient Boosting
Machine), KNN (K-Nearest Neighbor), RF (Random Forest), SVM (Support Vector
Machine), NSE (Nash-Sutcliffe efficiency), $R^2$ (Coefficient of Determination), Qp (The
relative error of flood peak flow), Tp (The absolute error in flood peak occurrence time),
VIF (Variance inflation factor), RMSE (Root mean square error), STD (Standard
deviation).

## 4.  Result

### 4.1. Model performance

The Top-SSF model demonstrated good flood simulation performance across the
80 gauged catchments, as quantified by NSE, Qp, and Tp. During the calibration period,
50% of the catchments achieved NSE values exceeding 0.78 (Fig. 5a), the median Qp
value was below 10% (Fig. 5b), and the median Tp value was within 2 hours (Fig. 5c).
The average NSE value was approximately 0.8, with a maximum of 0.96. The majority
of Qp values were around 8%, and the majority of Tp values were below 2 hours.
During the validation period, the median NSE value was 0.76 (Fig. 5a), the median Qp
value was below 10% (Fig. 5b), and the median Tp value was within 4 hours (Fig.5c).
The hydrological response times for the 80 catchments were approximated as the time
from precipitation peak to flood peak. The estimated range is from 1 to 26 hours. This
diversity is indicative of the comprehensive nature of the study, which encompasses
both rapid flash floods in smaller basins and more general floods in larger, mountainous
catchments (mean area: 1,586 km²). For catchments with longer response times, a
median error of 2-4 hours remains operationally valuable for providing sufficient flood
warning lead time. It is noteworthy that the median Tp during the calibration period
(within 2 hours) satisfied China's Specification for Hydrological Information Forecast
(GB/T 22482-2008) stringent requirements for high-quality forecasts.
Model performance also exhibited some dependence on catchment characteristics.
For instance, NSE generally improved with increasing forest cover (Fig. 6a), potentially
due to the model's explicit representation of forest canopy interception and subsurface
storm flow generation mechanisms. The relationship between NSE, Qp, Tp and
elevation was more complex, suggesting a nonlinear influence of elevation on model
performance (Fig. 6 a-c). The demonstrated robust performance of the Top-SSF model
provides a strong foundation for its application in subsequent parameter regionalization
analyses.

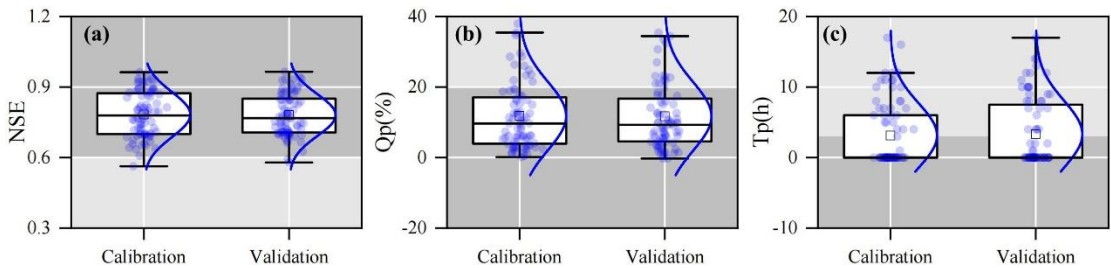

**Fig. 5.** Boxplots of (a) NSE, (b) Qp, and (c) Tp during the calibration and validation periods for 80 gauged catchments. The box represents the interquartile range, with the middle line indicating the median (50th percentile). The whiskers represent the minimum and maximum values. "□" represents the mean value. Dark grey indicates the range of flood prediction criteria (i.e., NSE> 0.75, Qp< 20%, and Tp < 2 hours).

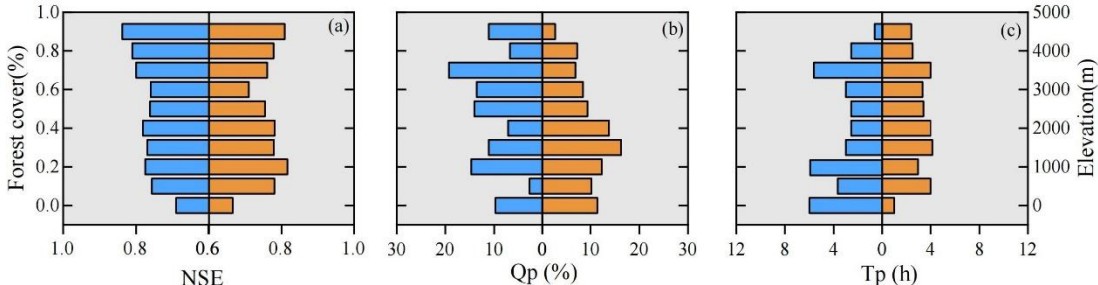

**Fig.6.** Influence of environmental factors on Top-SSF model performance in flood simulation. The graphs illustrate the relationship between model evaluation metrics and forest cover (left) and elevation (right).

## 4.2. Results of parameter regionalization

### 4.2.1. Comparison of sensitive model parameter estimates

The six single machine learning regionalization methods exhibited varying performance in estimating sensitive model parameters (Fig. 7), likely due to differences in catchment descriptor characteristics and the underlying principles of each method. Their hyperparameter results are presented in Tables S1–S6 of the supplementary material. The GBM demonstrated the highest accuracy in estimating $Szm$, $td$, and $C$ ($R^2$ = 0.90, 0.86, and 0.87, respectively,), with its estimates also exhibiting a STD that closely matched the distribution of the calibrated parameter values. KNN provided the most accurate estimates for $lnTe$, $qsf0$, and $t$ ($R^2$ = 0.87, 0.89, and 0.90, respectively), also with STD closely resembling the calibrated parameter distributions.

ERT performed best in estimating $Sfmax$ ($R^2$ = 0.87), but its performance was
generally poorer for other parameters. DT, SVM, and RF methods generally showed
lower performance across all sensitive model parameters. These differences in
performance highlight the potential benefits of multi-machine learning ensemble
methods for improving flood prediction in ungauged mountainous catchments.

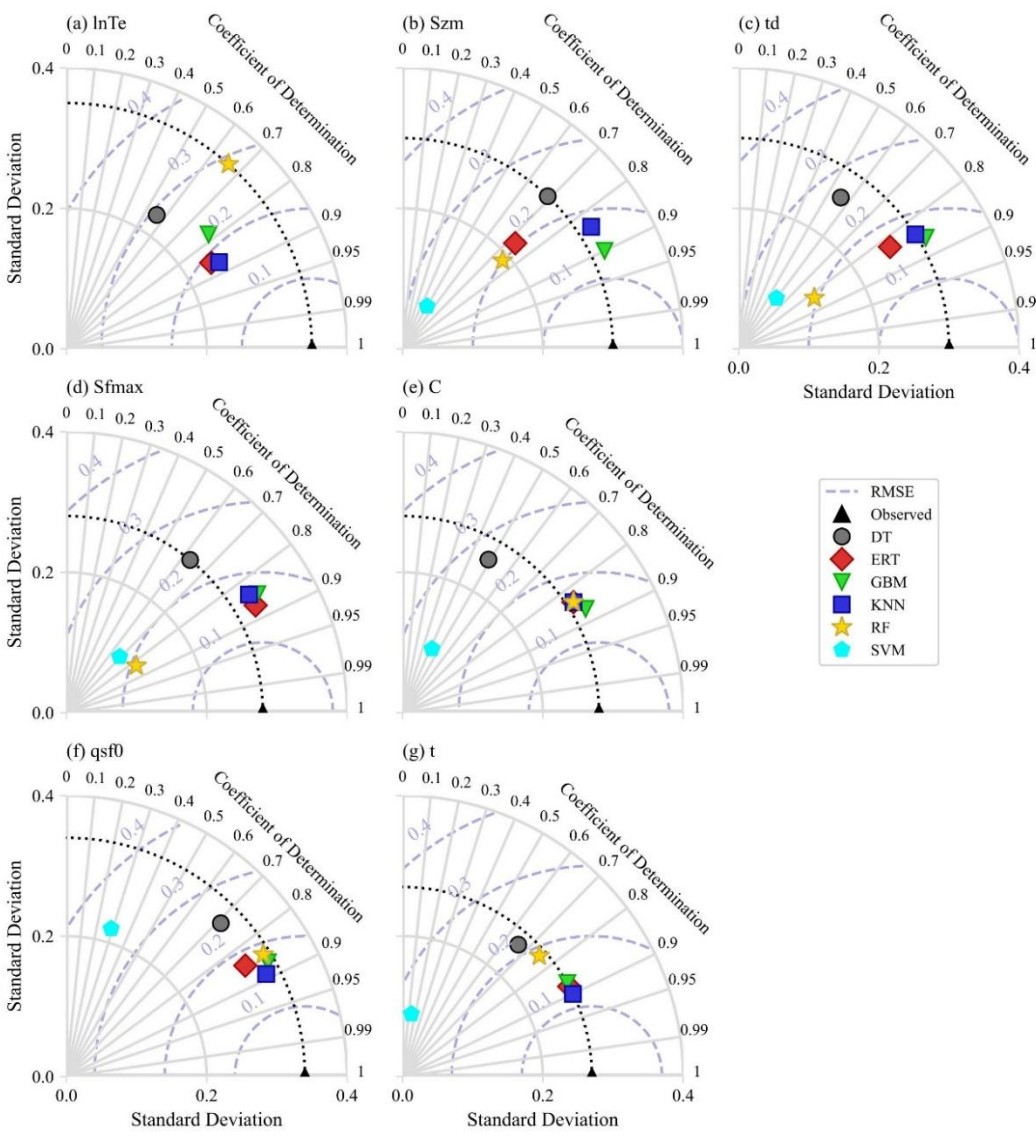


**Fig.7.** Performance of parameter regionalization methods assessed using Taylor diagrams. The
diagrams show the accuracy of sensitive model parameter estimates, with the coefficient
of determination ($R^2$) indicated by the radial axis, standard deviation (STD) by the
horizontal and vertical axes, root mean square error (RMSE) by the grey-blue dotted lines,
and the standard deviation of observations by the black dotted line.

**4.2.2. Comparison of flood forecasting results**

The flood prediction performance of the Top-SSF model, integrated with different parameter regionalization methods, was compared across 80 mountainous catchments in southwestern China. The methods included single machine learning methods and a multi-machine learning ensemble method (GBM-KNN-ERT), where GBM estimated $Szm$, $td$, and $C$; KNN estimated $lnTe$, $qsf0$, and $t$; and ERT estimated $Sfmax$. The performance of these parameter regionalization methods was then evaluated against the performance of the Top-SSF model using calibrated parameters. Among the single machine learning methods, GBM performed best, with 60 catchments achieving a positive NSE (NSE > 0, Fig. 8d). Critically, for high-accuracy predictions (NSE > 0.9), GBM succeeded in 43 catchments (54%), also showing strong performance with Qp less than 5% and Tp less than 1 hour in most cases (Fig. 8a-c). The GBM-KNN-ERT ensemble method yielded even better results. It increased the number of catchments with positive NSE to 75 (Fig. 8d). More impressively, the ensemble method achieved exceptional performance (NSE > 0.9) in 72 catchments (90%). This represents a 67.44% increase in the number of high-accuracy predictions compared to the best single method (GBM). Furthermore, the ensemble method Qp values were more concentrated around zero, and 90% of catchments maintained near-zero Tp values. These results strongly demonstrate the superior potential of multi-machine learning ensembles for improving flood prediction in ungauged catchments.

To further illustrate these performance differences visually, Fig. 8 (e, f, and g) presents hydrographs from three randomly selected flood events. These events

represent cases where the calibrated Top-SSF model itself achieved high (NSE=0.91),
medium (NSE=0.76), and low (NSE=0.55) performance, respectively. A key insight
from these plots is that the Top-SSF simulation (solid black line) is the performance
benchmark for the regionalization methods. Although the models aim to approximate
measured floods, their performance is ultimately limited by the accuracy of the Top-
SSF model structure and its optimized parameters.
The hydrographs show how the GBM-KNN-ERT ensemble achieves superior
performance by leveraging the complementary strengths of its component methods. For
instance, in the high-performance case (Fig. 8e), the GBM and KNN methods capture
the overall shape well, but the ERT simulation provides a more precise estimation of
the primary flood peak. The final ensemble successfully integrates this peak accuracy,
resulting in the highest overall performance. Similarly, Fig. 8f shows that the ensemble
moderates the slow initial rise characteristic of the KNN method, leading to a more
realistic rising limb. The ensemble method ability to balance competing errors is most
evident in the low-performance case (Fig. 8g). During the recession phase, the ensemble
method averages the high bias of the ERT method with the low bias of the GBM and
KNN methods, producing a hydrograph that more closely resembles the benchmark
simulation than any single model could. This synergy demonstrates that the ensemble
method superior performance is a direct result of its ability to integrate the specific,
complementary strengths of each member model across different parts of the
hydrological process.

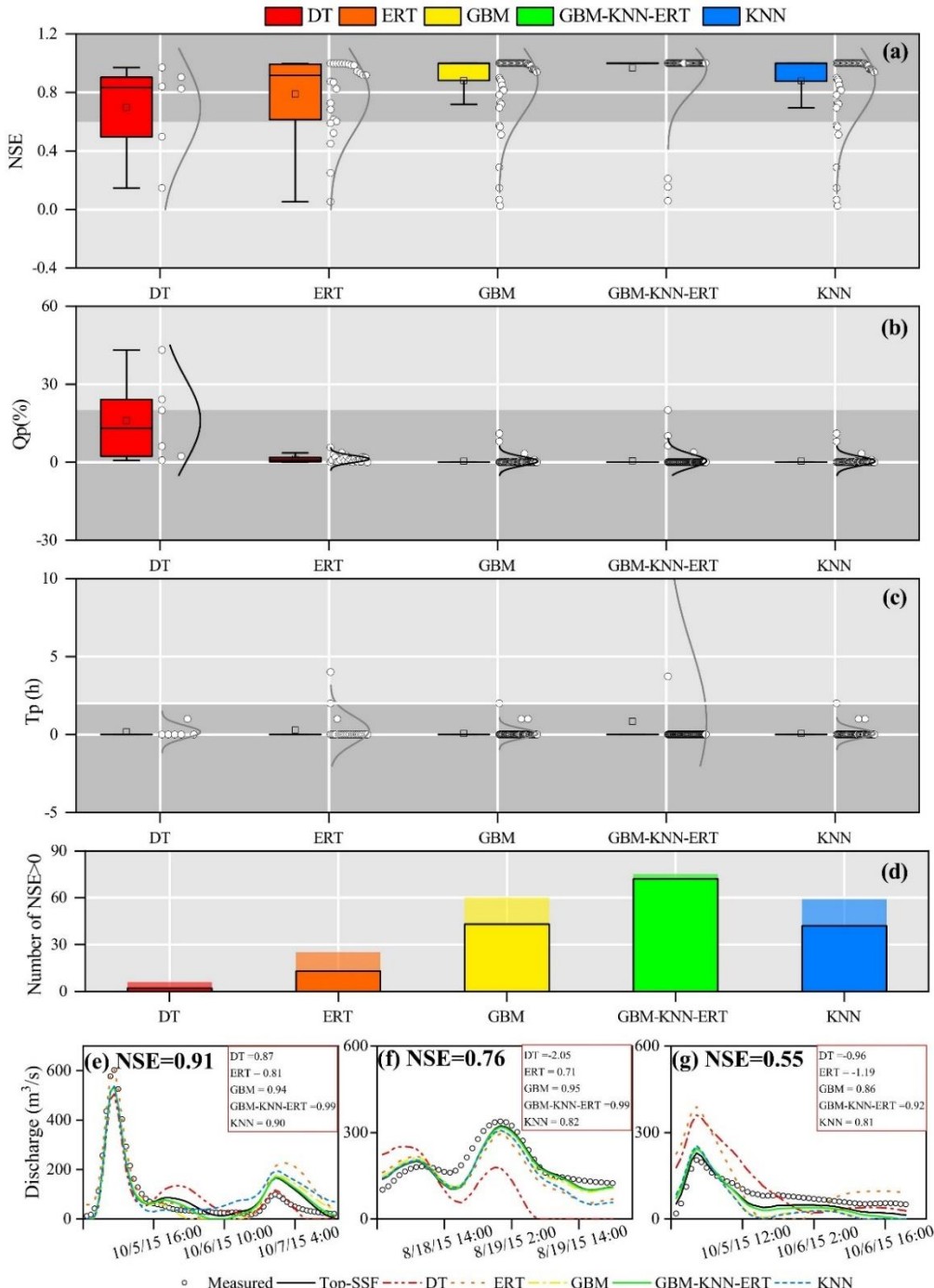


**Fig.8.** Evaluation of flood prediction performance for different parameter regionalization
methods. (a-c) show the distributions of Nash-Sutcliffe Efficiency (NSE), relative peak
flow error (Qp), and peak time error (Tp) across all 80 catchments, with shaded regions
indicating where flood prediction standards were met (NSE > 0.75, Qp < 20%, and Tp <
2 hours). (d) shows the number of catchments with NSE > 0 and the black border indicates
the number of catchments with NSE > 0.9. (e-g) present example hydrographs comparing
the simulated flood from each regionalization method against measured flood flow and
the calibrated Top-SSF model benchmark for catchments where the benchmark model
performance was (e) high (NSE=0.91), (f) medium (NSE=0.76), and (g) low (NSE=0.55).

## 5.  Discussion

### 5.1.  Reliability of multi-machine learning ensemble in parameter regionalization

In this study, the GBM-KNN-ERT method demonstrated superior regionalization performance, highlighting the potential of ensemble methods for improving hydrological predictions in ungauged mountainous catchments. The success of the ensemble is rooted in the distinct learning mechanisms and behaviors of its individual components, which were revealed during hyperparameter optimization.

The GBM method exhibited distinct parameter-specific sensitivities to hyperparameters (Fig. 9a-c). For parameter $C$, the negative correlation between $R^2$ and n_estimators (>300 trees) indicates overfitting risks when modeling complex rainfall-runoff interactions in heterogeneous mountainous terrain (Fig. 9a). This aligns with previous findings emphasizing the need for complexity control in hydrological generalization (Schoups et al., 2008). Conversely, the improved $R^2$ for parameter $td$ with increased n_estimators highlights the capacity of ensemble learning to capture complex, nonlinear relationships between catchment descriptors and hydrological parameters (Hastie et al., 2009). The contrasting optimal max_depth of 10 layers for parameter $C$, compared to shallower optimal depths (3-4 layers) for $Szm$ and $td$, suggests that parameters governing more complex hydrological processes in mountainous catchments may require deeper decision trees to effectively capture the interactions between climate, topography, and soil properties (Wainwright and Mulligan, 2013).

KNN performance exhibited pronounced sensitivity to neighbourhood size

(n_neighbors) and distance metric (p), highlighting the spatial heterogeneity of
catchment descriptors. For parameters $lnTe$ and $qsf0$, optimal performance was
observed at n_neighbors =30 (Fig. 9d), aligns with the hypothesis that meaningful
hydrological similarities can emerge even in topographically complex mountainous
regions when considered at broader spatial scales (Li et al., 2022). Conversely,
parameter $t$ achieved peak accuracy at n_neighbors=5, suggesting that localized,
short-term weather events and fine-scale topographic similarities in adjacent
mountainous areas can significantly influence local runoff processes (Garambois et al.,
2015). The Manhattan distance metric (p=1) outperformed Euclidean distance across
all parameters (Fig. 9e). This performance advantage is primarily attributed to the
method's capacity to alleviate the "curse of dimensionality" (Bellman, 1961) inherent
in high-dimensional datasets—a prevalent challenge when characterizing complex
mountainous catchments with diverse descriptors. In such datasets, sparse data
distributions and the presence of mixed variable types (e.g., topographic indices, land
cover) can significantly degrade the discriminative power of Euclidean distance
(Rockström et al., 2023). The robustness of the Manhattan distance arises from its axis-
aligned sensitivity, which provides a more effective means of handling feature scaling
and integrating catchment descriptors compared to the radial symmetry of Euclidean
distance.

ERT performance was maximized at max_features = 0.1 (Fig. 9f). By restricting

the random sampling of features during node splits (using only 10% of the features),
both the diversity of the trees was enhanced and the effects of multicollinearity between
topographic and soil attributes were reduced. This finding aligns with the theory
proposed by Geurts et al. (2006), which suggests that random feature selection can
significantly improve model generalization, a particularly important consideration in
ungauged mountainous catchments characterized by high levels of inter-correlation
among predictor variables.

These distinct sensitivities and learning mechanisms form the scientific basis for

the superiority of the GBM-KNN-ERT method. As shown in Section 4.2, no single
machine learning method is universally optimal for all hydrological model parameters.
Instead, the ensemble method effectively allocates each parameter to the model best
suited for its regionalization. Specifically, GBM, with its capacity for modeling
complex interactions, proved optimal for integrated parameters like $Szm$ and $td$. In
contrast, the instance-based KNN was superior for parameters like $lnTe$, which are
governed by physical similarity and spatial coherence. Finally, the highly randomized
nature of ERT provided the necessary robustness to model the noisy relationship
associated with the $Sfmax$ .This synergistic combination, where each model
contributes its unique strength, results in a final regionalization framework that is more
accurate and physically plausible than any individual method operating in isolation.

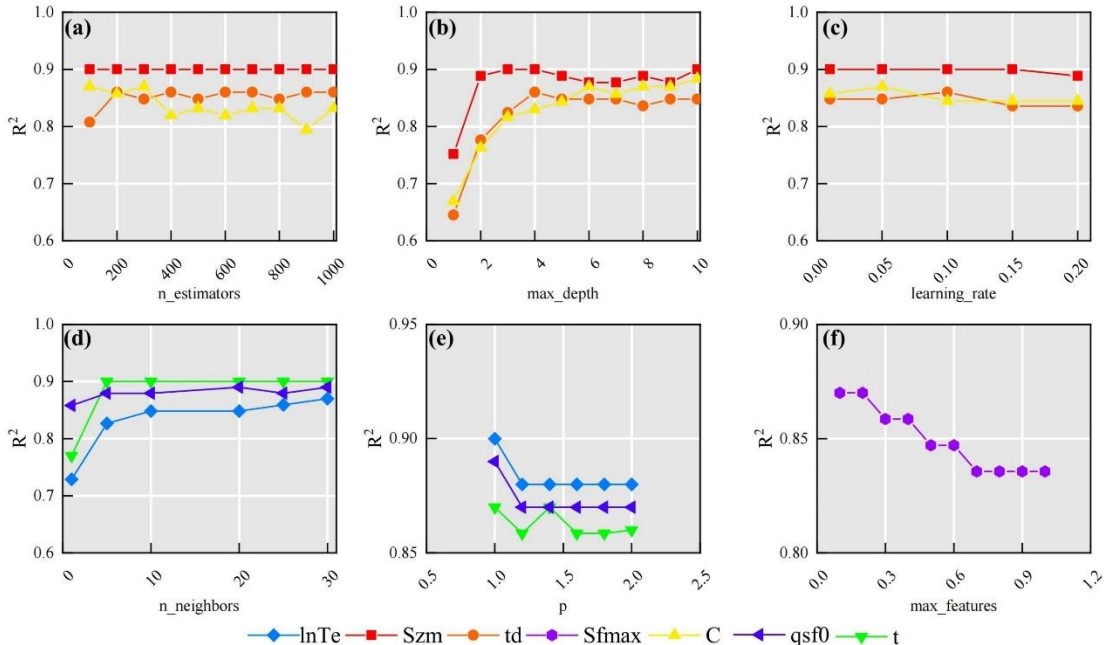

**Fig.9.** Sensitivity of parameter estimation performance to key hyperparameters in (a-c) GBM,
(d-e) KNN method, and (f) ERT. (a) n_estimators (number of decision trees in GBM), (b)
max_depth (maximum depth of decision trees in GBM), (c) learning rate (GBM), (d)
n_neighbors (number of neighbors in KNN), (e) p-value of Minkowski distance (KNN;
p=1: Manhattan distance, p=2: Euclidean distance), and (f) max_features (ERT).
**5.2. Combining multiple machine learning methods for parameter regionalization**
Machine learning methods exhibit distinct strengths in hydrological parameter
estimation due to fundamental differences in data processing mechanisms, pattern
recognition strategies, and prediction generation (Bishop and Nasrabadi, 2006). This
suggests that multi-machine learning ensemble methods have the potential to
synergistically integrate advantages while effectively compensating for individual
limitations, leading to more robust and accurate parameter estimates. As demonstrated
in Fig. 10, the GBM-KNN-ERT method achieved notable improvements over any
single machine learning method, particularly for sensitive parameters $lnTe$, $Sfmax$,
$qsf0$ and $t$, with $R^2$ increases ranging from 0.02 to 0.03 compared to the best-
performing GBM method (Fig.10e).
Interestingly, a comparison of GBM4-KNN3 (where $Sfmax$ is estimated by
GBM) and GBM3-KNN4 (where $Sfmax$ is estimated by KNN) revealed critical
insights into model parameter compatibility. Despite both achieving an identical $R^2$ of
0.85 for the estimation of $Sfmax$, GBM4-KNN3 exhibited superior flood prediction
performance, with 72 catchments achieving NSE > 0 compared to only 68 catchments
for GBM3-KNN4. This suggests that GBM possesses an enhanced capability to resolve
the complex coupling between soil moisture dynamics and topography, leading to more
physically plausible representation of subsurface storm flow processes (Gupta et al.,
2023). The wider distribution of flood prediction performance observed for GBM3-
KNN4 (Fig. 10 a–c) further suggests that uncertainties introduced by KNN in the
estimation of $Sfmax$ may propagate nonlinearly during flood simulations, potentially
amplifying errors. This observation aligns with theoretical expectations that distance-
based methods may tend to oversmooth critical thresholds or sharp transitions in
heterogeneous environments, leading to a less accurate representation of hydrological
responses (Bellman, 1961).
Furthermore, an important consideration in adopting ensemble methods is the
trade-off between predictive accuracy and computational efficiency. To evaluate this
trade-off, we compared the model training times for various parameter regionalization
methods, with the results summarized in Table 4. The analysis shows that our proposed
GBM-KNN-ERT ensemble, while providing the highest predictive accuracy, required
a total training time of 102.8 s. This is moderately higher than the best-performing
single model, GBM (57.6 s), and other simpler ensemble methods like GBM4-KNN3
(36.1 s). The increased computational time for the GBM-KNN-ERT method is
primarily attributed to the inclusion of the ERT method for estimating the $Sfmax$,
which is inherently more computationally intensive than GBM or KNN.
However, it is crucial to contextualize this computational cost for operational use.
The process of training a regionalization method is an offline task, performed once to
establish the stable relationships between catchment descriptors and model parameters.
This one-time investment is not a constraint on real-time flood forecasting, as once the
method is trained, parameter estimation for a new ungauged catchment is nearly
instantaneous. To provide context for the reported computational times, all model
training and simulations were performed on a workstation equipped with an Intel(R)
Core (TM) i9-10900K CPU @ 3.70GHz, 32.0 GB of RAM, and an NVIDIA Quadro
P1000 (4 GB) GPU, running on a 64-bit Windows operating system with Python 3.9.
Given this context, the modest increase in one-time training cost is a justifiable
investment for the significant improvements achieved in flood prediction accuracy,
model robustness, and stability. Therefore, for applications in water resource
management and flood risk assessment where high accuracy is paramount, the GBM-
KNN-ERT method strikes an optimal and practical balance between computational
efficiency and predictive performance.

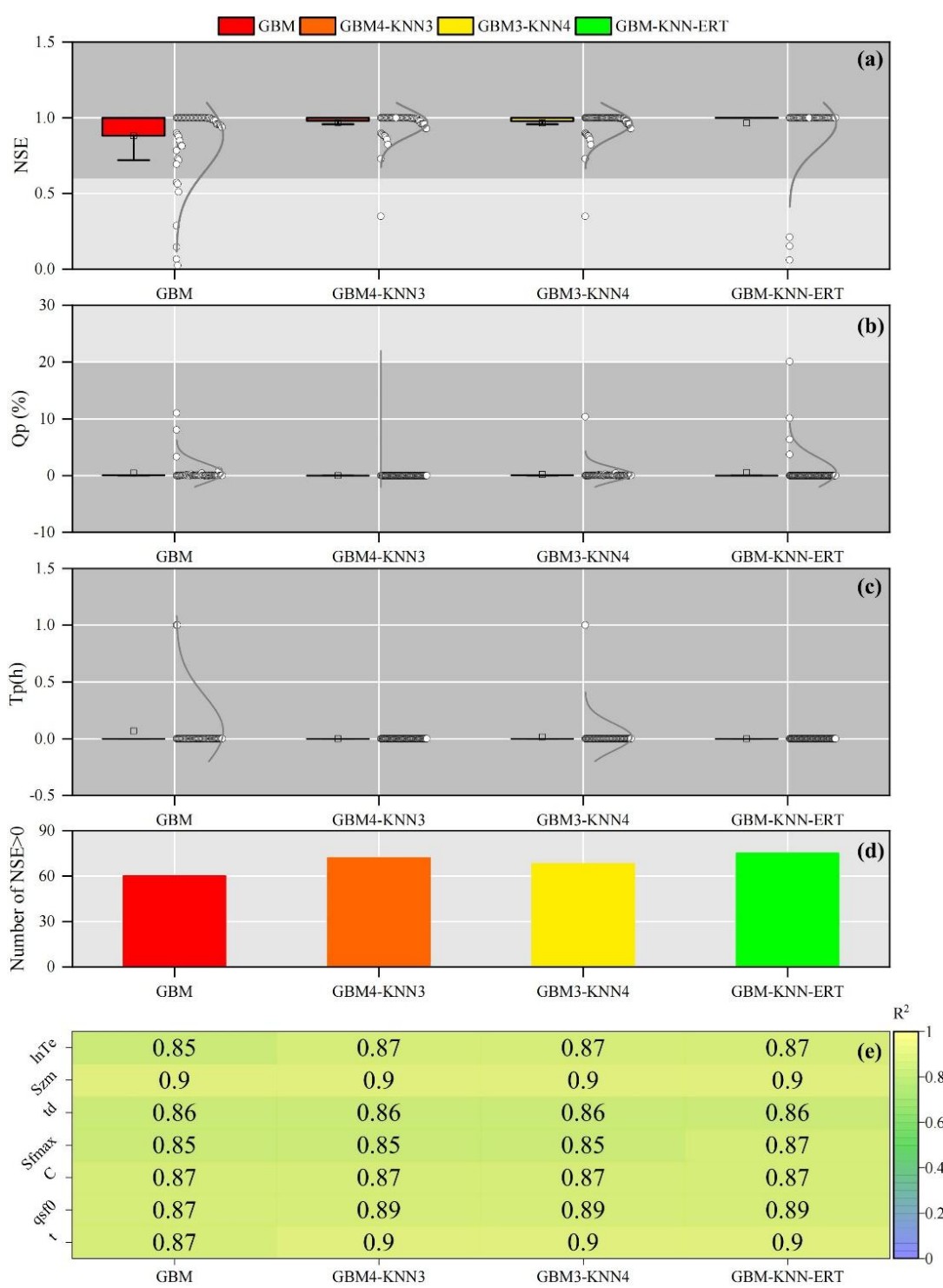


**Fig.10.** Assessment of combined machine learning methods for improved parameter
regionalization in ungauged mountainous catchments. Performance is evaluated against
the GBM method, showing (a) NSE, (b) Qp, (c) Tp, (d) Number of catchments with NSE >
0, and (e) the difference in $R^2$.





**Table 4.** Running time (s) for different parameter regionalization methods

| | GBM | GBM4-KNN3 | GBM3-KNN4 | GBM-KNN-ERT | KNN | ERT |
|---|---|---|---|---|---|---|
| $lnTe$ | 11.3 | 3.4 | 3.4 | 3.7 | 3.6 | 74.4 |
| $Szm$ | 7.8 | 7.5 | 7.7 | 7.8 | 0.6 | 76.7 |
| $td$ | 8.2 | 8.1 | 8.0 | 8.5 | 0.6 | 74.7 |
| $Sfmax$ | 7.7 | 8.2 | 0.6 | 73.6 | 0.5 | 74.9 |
| $C$ | 7.8 | 7.7 | 7.7 | 8.0 | 0.6 | 74.9 |
| $qsf0$ | 7.4 | 0.6 | 0.6 | 0.6 | 0.6 | 76.3 |
| $t$ | 7.4 | 0.6 | 0.6 | 0.6 | 0.5 | 75.3 |
| Sum | 57.6 | 36.1 | 28.6 | 102.8 | 7.0 | 527.2 |

**5.3. The influence of donor catchment quantity on machine-learning parameter**
**regionalization**

575   The number of donor catchments used in machine learning-based parameter

regionalization methods is a critical factor influencing the accuracy and robustness of
hydrological predictions in ungauged catchments (Gauch et al., 2021; Song et al., 2022;
Zhang et al., 2022). In this study, we investigated the influence of donor catchment
quantity (ranging from 20 to 80) on the flood prediction performance of the two best-
performing parameter regionalization methods (GBM4-KNN3 and GBM-KNN-ERT)
across the 80 mountainous catchments (Fig 11). It is important to clarify that the
following analysis is not a method for selecting donor catchments based on physical
similarity—a task handled by the machine learning methods itself when it learns the
relationships between catchment descriptors and model parameters. Instead, this
experiment serves as a sensitivity analysis to understand how the regionalization
performance is affected by the overall quantity and quality of the available training data.

587   To systematically investigate the performance influence of donor catchment

quantity on parameter regionalization, two distinct sampling strategies were employed
across the 80 mountainous catchments. In Mode 1 (selection of donor catchments based

on decreasing NSE), which was designed to test the impact of data quality, a non-monotonic relationship was observed. For both methods, regionalization performance peaked with 20-40 donor catchments and then declined, particularly for the GBM4-KNN3 method (Fig. 11a-c). This performance degradation is not due to increasing catchment dissimilarity, but rather to the introduction of lower-quality training data. As the donor pool expands beyond the best-performing catchments, it begins to include catchments where the Top-SSF model calibration itself was less successful (i.e., lower NSE values). These 'low-quality' samples may introduce noise and less reliable parameter-descriptor relationships, which can mislead the training process (Gauch et al., 2021; Zhang et al., 2022). Notably, the GBM-KNN-ERT method demonstrated greater resilience to this degradation. Its performance, while also peaking early, did not degrade as sharply and instead tended to stabilize after the inclusion of approximately 70 catchments. This suggests that the more complex ensemble structure has a superior ability to suppress noise and generalize from a dataset containing a mix of high- and low-quality examples, highlighting its enhanced robustness. In contrast, Mode 2 (random selection of donor catchments) demonstrated a consistent improvement in regionalization performance for both NSE and Tp as the number of donor catchments increased (Fig. 11d-f). However, while the average performance improves with data quantity, it is important to acknowledge that this trend relies on the random samples being generally representative; a poorly chosen random set could still reduce generalizability. Notably, under both modes, the GBM-KNN-ERT method consistently exhibited significantly greater performance stability compared to the alternative

ensemble, GBM4-KNN3. This enhanced robustness likely arises from its more
effective suppression of data heterogeneity and noise interference, indicating that more
complex ensemble methods possess a greater capacity to balance the benefits of
increased data quantity with the potential drawbacks of reduced data quality.

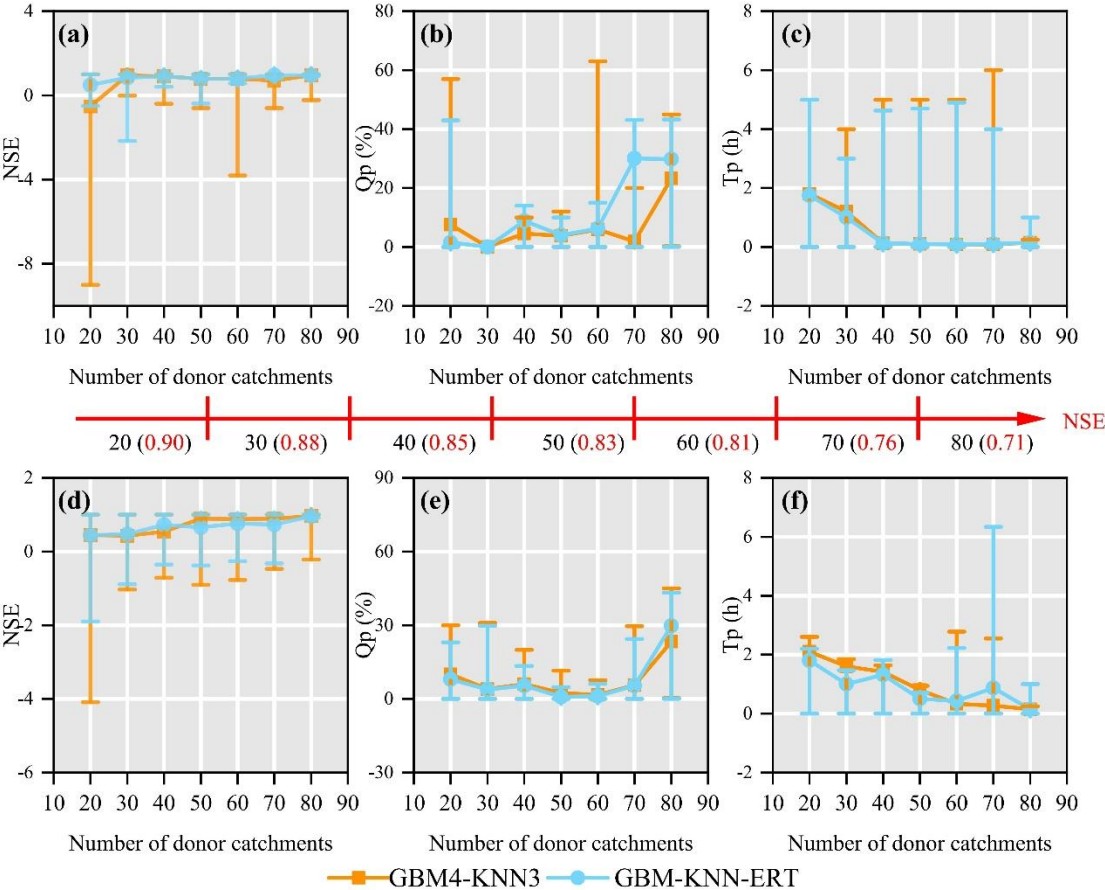

■ GBM4-KNN3 ● GBM-KNN-ERT
**Fig. 11.** Performance comparison of two donor catchment selection methods for parameter
regionalization as a function of donor catchment quantity. Mode1 (a-c) selects donor
catchments in order of decreasing NSE, while Mode 2 (d-f) selects them randomly. Flood
prediction accuracy is assessed using NSE, Qp, and Tp. Error bars represent the full range
(minimum to maximum) of the performance metrics.
**5.4. The impact of climate change on parameter regionalization methods**
The hydrological cycle within catchments is fundamentally governed by complex
interactions between climate and environmental factors. The Intergovernmental Panel
on Climate Change (IPCC) has consistently documented a continuous and accelerating
transition in global climatic patterns, characterized by increased variability and extreme
events (Pachauri et al., 2014). Consequently, future flood predictions derived from
parameter regionalization methods are expected to exhibit increased uncertainty and
variability, highlighting the substantial influence of climate change on the reliability
and precision of flood predictions in ungauged mountainous catchments (Yang et al.,
2019). Therefore, a sensitivity analysis was designed to evaluate the robustness of the
trained regionalization models when confronted with climatic conditions outside their
original training range.
To quantitatively assess the impact of climate change, an experiment was devised
where this impact was primarily reflected through changes in two key catchment
descriptors: Tem and Pre. For the historical period, these descriptors represent the multi-
year averages over 1901–2021, while for the future period, they represent the projected
multi-year averages over 2022–2100 under the SSP5-8.5 scenario. The regionalization
methods (GBM4-KNN3 and GBM-KNN-ERT), which were trained exclusively using
historical data, were then applied under these future conditions. Crucially, the method
structures and hyperparameters remained fixed, and no retraining was performed; only
the historical Tem and Pre values were replaced with their future projections. This
approach allows the response of the established historical relationships to new, out-of-
sample climatic inputs to be tested. The simulated peak discharges for this analysis were
derived from the same three flood events used in the calibration and validation of the
Top-SSF model. This experimental design is critical as it isolates the impact of the
changed model parameters from the compounding effect of a different future rainfall
event. Consequently, any observed change in the simulated flood peak is attributable

solely to the sensitivity of the regionalization method to the shift in climatic descriptors. Cumulative distribution functions (CDFs) were then employed to illustrate the discrepancies between the parameter regionalization simulations and the reference simulations (derived from calibrated model parameters) across the historical and projected future periods for the 80 catchments (Fig.12).

A comparative analysis of Fig. 12a and 12b reveals a clear amplification of the absolute differences in predicted flood peaks (quantified as the error in runoff modulus) between the two parameter regionalization methods and the reference Top-SSF model simulations during the transition from the historical period to the projected future period. Specifically, the maximum error in runoff modulus for the GBM4-KNN3 method increased by 68.46 $m^3$ $s^{-1}$ $km^{-2}$ from the historical period to the future period, while the increase for the GBM-KNN-ERT method was a smaller 56.65 $m^3$ $s^{-1}$ $km^{-2}$. These results underscore that parameter regionalization methods are inherently sensitive to changing climatic forcing. However, they also provide compelling evidence that the GBM-KNN-ERT method exhibits superior stability and resilience under climate change, demonstrating its potential for more reliable long-term flood risk assessment in ungauged mountainous regions.

Exploring the effects of climate change on parameter regionalization methods provides valuable insights for advancing flood prediction research in prediction in ungauged basins. The enhanced stability demonstrated by the GBM-KNN-ERT ensemble offers a promising direction for developing robust regionalization methods capable of navigating the challenges of a non-stationary climate.

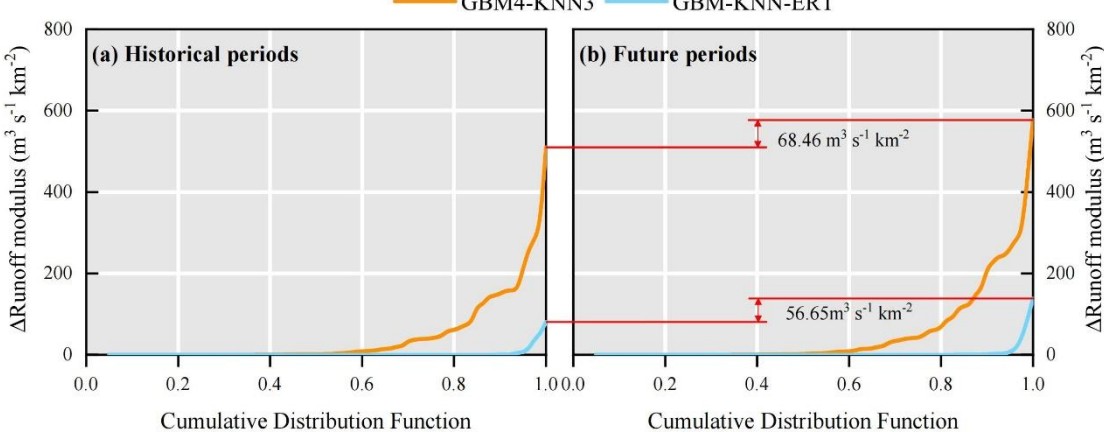

**Fig.12.** Comparison of flood peak runoff modulus between parameter regionalization and
calibrated Top-SSF model results, showing cumulative distribution functions (CDFs) of
absolute differences for 80 catchments during (a) historical and (b) future periods.
**5.5. Uncertainty and limitation**
The uncertainty in this study arises from several sources, including the
hydrological model, the regionalization methods, and the data itself. A critical
evaluation of these sources helps to contextualize our findings and assess the
generalizability of the ensemble method. Uncertainty from the hydrological model is
inherent in its structure and the calibrated parameters. Although the Top-SSF model
performed well, its parameters are effective values subject to equifinality. This
uncertainty in the "true" parameter values can be viewed as a form of calibration bias,
which serves as the target data for our regionalization. To mitigate this, we employed
the robust SCE-UA optimization algorithm and focused only on sensitive parameters.
Uncertainty is also introduced by the regionalization methods themselves, as the
training data derived from donor catchments are susceptible to errors that can impact
model performance (Mosavi et al., 2018; Xu and Liang, 2021).
A specific methodological choice was the exclusion of deep learning architectures,
such as Multilayer Perceptrons or Long Short-Term Memory (LSTM) networks. This
decision was guided by several factors. First, parameter regionalization is a static
regression problem, mapping time-invariant catchment descriptors to model parameters,
which does not align with the sequential data structure for which LSTM is designed.
Second, deep networks typically require large datasets to avoid overfitting; with a
dataset of 80 catchments, traditional machine learning methods like GBM and ERT are
often more robust and less prone to memorizing training data. Third, a key advantage
of parameter regionalization is its potential for physical interpretability. Unlike DL
models, whose internal decision-making processes are often obscured within abstract
weight matrices, the ensemble methods employed here offer more accessible
transparency. The tree-based models (GBM and ERT) allow for the direct assessment
of feature importance, enabling the verification of physical consistency. Furthermore,
the KNN component provides "instance-based" interpretability by explicitly identifying
the specific donor catchments used for transfer. This preserves the traceable logic of
hydrological similarity, clearly indicating the geographical or physical source of the
transferred parameters, a level of insight that is crucial for building trust in water
resource management.
Furthermore, the primary contribution of this study is not the identification of a
single superior algorithm, but the demonstration of a data-driven framework for
constructing a locally optimal ensemble. The complementarity of the chosen models
was not assumed but empirically validated through a competitive evaluation process.
Each of the seven machine learning methods was independently trained and assessed
for its ability to estimate each sensitive parameter. The final GBM-KNN-ERT ensemble
was constructed by selecting only the empirically best-performing model for each
parameter based on objective metrics ($R^2$, RMSE, STD). The very fact that different
methods were selected for different hydrological parameters provides direct empirical
evidence of their complementary strengths, thus validating the ensemble method.

Furthermore, the specific GBM-KNN-ERT combination identified is necessarily

data-dependent, raising questions about its transferability. However, this study primary
contribution is not the specific model combination itself, but rather the demonstration
of a data-driven method for constructing a locally optimal ensemble. This method is
designed to be generalizable; applying the same competitive evaluation process to a
new region would identify the best ensemble for that specific dataset. The key to
overcoming these limitations and ensuring robust generalization lies in genuine model
complementarity. The ensemble method's success is not an artifact of overfitting to
calibration bias or data quirks. Instead, it stems from a physically plausible "division of
labor", where different models are empirically shown to be better suited for
regionalizing parameters governed by distinct physical processes. The ensemble
method's superior stability in the out-of-sample climate change stress test further
supports this conclusion, indicating that it has captured robust underlying relationships,
not just noise.

To manage methodological uncertainty, we employed K-fold cross-validation to

ensure robust performance evaluation and RandomizedSearchCV for hyperparameter
tuning to minimize overfitting (Bergstra and Bengio, 2012). A key methodological
decision was to evaluate the regionalization methods against the outputs of the
calibrated Top-SSF model, rather than directly against observed flood events. This
approach was chosen for two primary reasons. First, it isolates the performance of the
parameter regionalization itself. The calibrated simulation represents the theoretical
'best-case' performance for the given hydrological model structure; consequently, any
deviation from this benchmark can be directly attributed to imperfections in the
regionalization method, rather than being confounded by the inherent structural
limitations of the Top-SSF model. Second, this strategy ensures that the machine
learning models learn the underlying physical relationships intended by the
hydrological model, not simply mimic data noise or measurement errors present in the
observations. If trained against raw observations, the machine learning methods might
derive 'spurious' parameter sets that compensate for both the hydrological model's
structural flaws and observational errors. Such parameters could appear effective but
would lack physical meaning and generalizability. These measures, combined with the
evidence for model complementarity, provide a strong basis for the scientific validity
and potential for generalization of our proposed ensemble method.
**6.  Conclusions**
This study introduces a novel multi-machine learning ensemble method (GBM-
KNN-ERT) to enhance model parameter transferability and improve flood prediction
in ungauged mountainous catchments. The proposed GBM-KNN-ERT method
demonstrated a substantial advancement in both flood prediction accuracy and model
robustness, achieving exceptional performance with 90% of ungauged catchments
exhibiting a NSE exceeding 0.9, a significant 67.44% improvement compared to the

best single machine learning method evaluated in this study. Importantly, the GBM-KNN-ERT method exhibited remarkable stability under simulated climate change, thereby highlighting its potential for reliable application in non-stationary hydrological environments. Furthermore, the method demonstrated notable adaptability to varying donor-catchment configurations, where an optimal balance between predictive accuracy and computational efficiency with a relatively limited set of 20–40 high-quality donor catchments (NSE >0.85). By integrating the diverse strengths of multiple machine learning with hydrological model, the proposed methodology significantly advances the field of flood prediction in ungauged catchments, offering a reliable tool for water resource management and flood disaster mitigation.

## Acknowledgements

This research was supported by the Joint Funds of the National Natural Science Foundation of China (**U2240226**), the National Natural Science Foundation of China (**42271038**) and the National Key Research and Development Program of China (**2022FY100205**).

## Competing interests

The authors declare that they have no known competing financial interests or personal relationships that could have appeared to influence the work reported in this paper.

## Author contributions

In this study, K L, G W, and J G were responsible for the conceptualization of the research. Data curation was carried out by K L, L G, and X S, while formal analysis

was performed by K L, J G, and J M. The methodology was developed by K L, L G, P
H, and J L. Project administration was overseen by G W and J G. K L took the lead in
writing the original draft, and the writing, review, and editing process involved
contributions from K L, G W, J L, P H, J M, X Z, and J G.
**Code and data availability**
The code used in this study is available upon request from the authors. The
meteorological, soil characteristics, and topography datasets are publicly accessible
online, as detailed in Table 1. The hourly flood data for the 80 catchments were sourced
from China's Hydrological Yearbook. These data are not publicly available due to
governmental restrictions but can be accessed by contacting the corresponding author
for further information.

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
