# Peer review of "Multi-Machine Learning Ensemble Regionalization of Hydrological"

_EGUsphere, 2025_

## Author Comment (AC1)

Comments on manuscript entitled 'Multi-Machine Learning Ensemble Regionalization of Hydrological Parameters for Enhances Flood Prediction in Ungauged Mountainous Catchments' by Li et al.

The manuscript deals with developing a multi-machine learning ensemble method for regionalization of a hydrologic model (Top-SSF) over 80 catchments in southwestern China. The authors showed the improvement in performance using multi-machine learning method over single methods. While the manuscript is well-structured and results are clearly presented, there are some points need to be addressed before the publication of the manuscript. Please find the comments as follows:

**Response: Many thanks for your comments.**

1) Line 107: what's the range for catchments area?

**Response: Added 'ranging from 109 to 6564 km' in Section2.1.**

2) Legend of Figure 1: please use the term 'Hydrometry station'

**Response: Revised**

3) Line 122: Hourly flow data

**Response: Revised**

4) Line 150: TOPMODEL not TOPMODE

**Response: Revised**

5) Section 3.1: More details should be provided. For example: What kind of hydrologic model is Top-SSF? Continuous or event-based? Lumped or (semi)distributed? And how it is going to be applied in this research? To simulate flood events? Or a whole time series (continuous modelling)? What are the inputs to the model, e.g. precipitation and temperature data?

**Response: More details of the Top-SSF model have been added in Section 3.1 and Section 3.3.1 of the revised manuscript.**

**Top-SSF is a semi-distributed hydrological model based on the well-established TOPMODEL framework, which delineates sub-basins based on the topographic index. It retains the key advantages of TOPMODEL, such as its parsimonious structure, physical interpretability, and ease of parameter transfer. In this study, while the model was driven by the continuous hourly meteorological data (including precipitation, temperature, surface pressure, relative humidity, wind speed, and net solar radiation), it was applied in an event-based manner to specifically simulate flood events. For each catchment, the model was calibrated using two independent, representative flood events and validated against a third, distinct flood event.**

6) Result section, Lines 362-365: Why performance of the different machine learning methods for parameter regionalization is compared against the Top-SSF model and not against the observed flood events?

**Response: The experiment was intentionally designed to compare the results of the machine learning methods with that of the Top-SSF model, isolating the performance of the parameter regionalization method themselves. To clarify this rationale, we have added a supplementary discussion in Section 5.5 of the revised manuscript. This method is based on two primary justifications:**

**First, the fundamental aim of the parameter regionalization is to effectively transfer model parameters to ungauged catchments, not to reconstruct or alter the model's underlying structure. By using the calibrated Top-SSF simulation as the benchmark, the theoretical "best-case" performance for that specific model structure was established. Consequently, any performance degradation observed in the regionalized models can be directly and exclusively attributed to the defects of the regionalization method, rather than being confounded by the inherent structural limitations of the hydrological model.**

**Second, this method ensures that we are assessing the regionalization method's ability to learn the underlying model physics, not to mimic data noise. While the Top-SSF model is calibrated against observed data (which is subject to measurement uncertainty), its output is a structurally consistent representation based on its physical equations. If we were to use the raw observations as the target, the machine learning methods might derive "spurious" parameter sets that compensate for both the hydrological model's structural errors and the observational errors. Such parameters might appear effective but lack physical meaning and generalizability. By targeting the Top-SSF simulation, we force the ML methods to learn the intended relationship between catchment attributes and the model's parameters, leading to a more robust and physically interpretable assessment of the regionalization techniques.**

7) Figures 11a and d: how can the NSE be greater than 1?

Response: **It is absolutely correct that the Nash–Sutcliffe Efficiency (NSE) has a theoretical upper bound of 1 which represents a perfect model simulation. However, in our initial visualization of Figures 11a and d, we used standard deviation to construct error bars for NSE, Qp, and Tp. It can be misleading for NSE due to its bounded nature (ranging from $-\infty$ to 1). As a result, the error bars erroneously extended beyond the physical limit of NSE = 1. To address this issue and ensure the accuracy of uncertainty representation, we have revised the calculation method for the error bars of NSE. Instead of using standard deviation, we now use the range (i.e., the difference between the maximum and minimum values) across the donor catchment configurations under each scenario. This revision ensures that the error representation remains within the theoretical bounds of NSE while still reflecting the variability of the model performance across the different donor catchment selections.**

**This correction improves the clarity and scientific validity of our results presentation without altering the main findings of the study.**

[Figure]

**Fig. 11.** Performance comparison of two donor catchment selection methods for parameter regionalization as a function of donor catchment quantity. Mode1 (a-c) selects donor catchments in order of decreasing NSE, while Mode 2 (d-f) selects them randomly. Flood prediction accuracy is assessed using NSE, Qp, and Tp. Error bars represent the full range (minimum to maximum) of the performance metrics.

8) Section 5.4: Not clear how the calculations carried out to simulate peak discharges. Which events in future are selected for this analysis? Did the whole time series of projected precipitation in baseline and future periods fed to the hydrologic model? Or just a few storms selected?

**Response: Yes, this is not clear. Clarifications have been added in this section. Specifically, in this part of the study, the impact of climate change is reflected through the changes in two catchment descriptors, i.e., mean annual temperature (Tem) and mean annual precipitation (Prec). Specifically, for the historical period, Tem and Prec represent the multi-year averages over 1901–2021; while for the future period, they represent the projected multi-year averages over 2022–2100 under the SSP5-8.5 scenario. To assess the influence of these climatic changes on flood prediction performance, we applied the parameter regionalization models (GBM4-KNN3 and GBM-KNN-ERT) calibrated by using historical data to future conditions. Under the unchanged model structures and hyperparameters, only the historical Tem and Prec values were replaced with their corresponding future projections. The simulated peak discharges were derived from the three flood**

events used in the calibration and validation of the Top-SSF model. We then compared the maximum flood peak discharge across all simulations between the historical and future periods to evaluate the absolute differences in runoff modulus.

This approach allowed us to isolate the effects of projected climate change on the stability and robustness of the parameter regionalization methods, particularly focusing on how changes in temperature and precipitation patterns influence flood peak predictions in the ungauged mountainous catchments.

---

## Author Comment (AC2)

**Overall evaluation**

The manuscript addresses an important problem: improving flood prediction in ungauged mountainous catchments through machine learning (ML) regionalization of hydrological parameters. The work has merit, particularly in its effort to combine multiple ML models into an ensemble and test the approach under a climate change scenario. That said, several aspects require clarification and expansion before the manuscript can be recommended for publication. In particular, the rationale for model choice, the justification for ensemble performance, interpretability of results, and methodological transparency needs to be strengthened. The manuscript would benefit from stronger connections between the ML methodology and hydrological processes.

**Response: Many thanks for your comments, which have significantly improved the manuscript. In response, we have substantially revised the paper to strengthen the rationale for our model selection and provide a clear scientific justification for the ensemble's superior performance. We now explain that its success stems from the complementary strengths of individual models, which are better suited to regionalizing parameters governed by distinct hydrological processes.**

**Methodological transparency and reproducibility have been enhanced by providing detailed hyperparameters, clarifying the climate change stability analysis, adding a discussion on computational efficiency, and including new hydrograph visualizations. These revisions create a much stronger and more interpretable link between our machine learning framework and the underlying physical hydrology, addressing the core concerns raised.**

**Specific comments**

1. The selected ML methods represent different learning paradigms (tree-based, instance-based, etc.). However, more complex techniques such as multilayer perceptron or deep learning networks were not included. The authors should justify why these were excluded and explain how they ensure a fair complementary across models that learn from very different principles.

**Response: Our selection of machine learning methods was deliberate, aimed at constructing a robust, interpretable, and computationally efficient regionalization framework. We carefully considered the role of deep learning models and designed our methodology to ensure the chosen algorithms are genuinely complementary and empirically validated.**

**(1) Justification for Excluding Deep Learning (DL) Models**

**Our decision to exclude deep learning architectures was based on several key factors, beginning with the fundamental nature of the research problem. A primary consideration is the distinction between our static regression task and the sequential nature of DL models like Long Short-Term Memory (LSTM). LSTMs are expertly designed for time-series data where temporal dependencies are critical. Our regionalization task, however, involves mapping a set of time-invariant catchment descriptors (e.g., area, slope) to time-invariant model parameters. Since there is no sequential relationship among these input features, applying an LSTM would represent a methodological mismatch, as its core**

**strength would be irrelevant.**

**While a more suitable deep learning architecture like a Multilayer Perceptron**
**(MLP) could be applied, such models are known to require large datasets to avoid**
**overfitting. With 80 catchments, our dataset is of a modest size for training deep**
**networks, posing a significant risk that an MLP would memorize the training data**
**rather than generalize to new, ungauged catchments. In contrast, ML methods like**
**GBM and ERT are well-established for their robust performance on structured,**
**tabular datasets of this scale.**

**Beyond these technical considerations, our decision was also guided by the**
**practical goals of our study. A key advantage of parameter regionalization over**
**purely data-driven forecasting is its enhanced physical interpretability. The tree-**
**based models we employed offer a degree of transparency, allowing for insights**
**into the relationships between catchment descriptors and hydrological parameters.**
**DL models, however, often function as "black boxes" making it challenging to**
**decipher the physical reasoning behind their predictions. As our aim was to**
**develop a transparent and reliable tool for water resource management,**
**maintaining this link to physical processes was a priority. Furthermore, the**
**computational cost of training and tuning DL models would have been**
**substantially higher, making our rigorous evaluation framework less feasible. On**
**the other hand, a novelty of our paper lies not in identifying the single most**
**powerful algorithm, but in demonstrating that an ensemble of diverse ML method**
**s can outperform any single ML method by leveraging complementary strengths.**

**(2) Ensuring Objective Complementarity Across Selected Models**

**To ensure this complementarity was achieved objectively, our methodology**
**was grounded in a systematic, data-driven process. Rather than pre-supposing**
**which model would be best for a given hydrological parameter, we firstly**
**conducted a comprehensive competition. Each of the seven ML methods was**
**independently trained and rigorously evaluated for its ability to estimate each of**
**the seven sensitive hydrological parameters, using K-fold cross-validation for**
**robust performance assessment. The final GBM-KNN-ERT ensemble was**
**constructed by selecting only the empirically best-performing model for each**
**specific parameter, based on the objective metrics ($R^2$, RMSE, STD). This**
**approach ensures that our final ensemble is not based on arbitrary choices but on**
**practical performance. The empirical outcome that different models were selected**
**for different hydrological parameters provides direct and powerful evidence of**
**their complementarity, validating our ensemble approach.**

**This detailed justification has been integrated into Section 5.5, "Uncertainty**
**and limitation".**

2.   The highlight statement that "The GBM-KNN-ERT method demonstrates superior
performance compared to other methods" is vague. Please clarify which performance
metrics are referred to, and quantify the magnitude of improvement.
**Response: It has been revised to be more specific and quantitative. The updated**

**highlight in the manuscript is as follows: 'The GBM-KNN-ERT method achieves high-accuracy flood predictions (NSE > 0.9) in 90% of catchments, a 67.44% improvement over the best-performing single ML method'.**

3.  The manuscript does not clearly explain how the trade-off between predictive accuracy and computational efficiency was considered. Given that ensemble methods can be computationally demanding, the authors should discuss the optimal balance and whether the proposed approach is practical for operational use.

**Response: We agree that a discussion on the trade-off between predictive accuracy and computational efficiency is essential, and we have added a new section to the manuscript to address this issue (Section 5.2). Our analysis of this trade-off is based on the running time of the ML methods (see the new Table 3 of the manuscript). This table shows that the GBM-KNN-ERT method, while being the most accurate, is indeed more computationally intensive than some of the single ML methods (such as KNN).**

**However, it is important to consider the computational cost in the context of its operational use. Training a ML method is an offline, one-time task performed to establish the relationships between catchment descriptors and model parameters before an application. It is not in the real-time forecasting workflow. Once the ensemble method is trained, estimating parameters for a new ungauged catchment is nearly instantaneous.**

**In this context, the computational overhead is minimal. A total training time of approximately 103s is highly practical, and importantly, the memory footprint of these ML method s is also low, enabling the entire workflow to be executed on standard computing hardware without requiring specialized resources. This low resource requirement ensures the method does not pose a significant barrier to implementation. The substantial gains in predictive accuracy, method robustness, and stability under climate change demonstrated throughout our paper far outweigh the modest, one-time increase in computational cost. Therefore, we conclude that the GBM-KNN-ERT method strikes an optimal and practical balance for applications where high accuracy is paramount, such as in water resource management and flood risk assessment. We have added this explanation to the end of Section 5.2 of the revised manuscript.**

**Table 3.** Running time ($s$) for different parameter regionalization methods

|        | GBM  | GBM4-KNN3 | GBM3-KNN4 | GBM-KNN-ERT | KNN  | ERT   |
|--------|------|-----------|-----------|-------------|------|-------|
| $lnTe$ | 11.3 | 3.4       | 3.4       | 3.7         | 3.6  | 74.4  |
| $Szm$  | 7.8  | 7.5       | 7.7       | 7.8         | 0.6  | 76.7  |
| $td$   | 8.2  | 8.1       | 8.0       | 8.5         | 0.6  | 74.7  |
| $Sfmax$| 7.7  | 8.2       | 0.6       | 73.6        | 0.5  | 74.9  |
| $C$    | 7.8  | 7.7       | 7.7       | 8.0         | 0.6  | 74.9  |
| $qsf0$ | 7.4  | 0.6       | 0.6       | 0.6         | 0.6  | 76.3  |
| $t$    | 7.4  | 0.6       | 0.6       | 0.6         | 0.5  | 75.3  |
| Sum    | 57.6 | 36.1      | 28.6      | 102.8       | 7.0  | 527.2 |

4.    The manuscript claims that the GBM-KNN-ERT method exhibits stability under climate change. However, the explanation of how climate change is incorporated is insufficient. While SSP585 projections (2022–2100) are mentioned, the methods used to integrate these into the ML framework and evaluate stability should be described in greater detail.

**Response: Yes, the methods were not described clearly enough. The primary goal of this analysis was not to forecast future floods, but to perform a sensitivity analysis on our trained regionalization methods (**GBM-KNN-ERT and GBM4-KNN3**) to evaluate their robustness when confronted with climatic conditions outside their original training range.**

**To address your concern, we have substantially revised Section 5.4 of the manuscript to provide a more detailed, step-by-step description of the methods. The key steps of this experimental design are as follows:**

**Step 1, the regionalization methods were trained exclusively using historical data. The climatic inputs for this training-mean annual temperature (Tem) and precipitation (Pre)- were based on the multi-year averages for the 1901-2021 period.**

**Step 2, a new input dataset was involved to represent future climatic conditions. For each catchment, we updated only the Tem and Pre descriptors with their projected multi-year averages for the 2022-2100 period under the SSP5-8.5 scenario. All other non-climatic descriptors remained unchanged.**

**Step 3, the already-trained regionalization methods were then used to predict a new set of Top-SSF parameters based on these future-conditioned inputs. Crucially, the regionalization method structures and hyperparameters remained fixed, and no retraining was performed. This approach isolates the response of the established historical relationships to new, out-of-sample climatic inputs.**

**Step 4, to evaluate the effect of these new parameters, they were input into the Top-SSF model to re-simulate the exact same historical flood events used during the original model calibration and validation. This design is critical as it isolates the impact of the changed model parameters from the compounding effect of a different future rainfall event. Consequently, any observed change in the simulated flood peak is attributable solely to the sensitivity of the regionalization method to the shift in climatic descriptors.**

**Step 5, regionalization method stability was quantified by comparing the absolute differences in simulated peak discharges between the reference simulations (using calibrated parameters) and the simulations using future-conditioned parameters, as illustrated by the cumulative distribution functions in Fig.12. A more stable regionalization method is one that exhibits a smaller increase in these differences when shifting from historical to future climate inputs.**

**This detailed process, which is now added in the revised manuscript (Section 5.4), allows us to robustly isolate and quantify the stability of our parameter regionalization methods against projected long-term shifts in climate.**

5. In the introduction, floods in mountainous catchments are mentioned, but it is unclear whether the focus is on general floods or flash floods (typically defined as occurring within 6 hours of rainfall). Given the rapid response of mountainous catchments, the authors should explicitly state which type of events are considered.

**Response: Yes, this is not clear. The Introduction and Section 2.2 (Datasets) have been revised to explicitly define the scope of the flood events.**

**In the Introduction: Floods in mountainous catchments, encompassing both flash floods and general larger-scale flood events which can be derived from mountainous upland catchments, pose a significant threat to human safety and property, particularly in regions lacking sufficient observational data.**

**In Section 2.2 (Datasets): Hourly flow data (2015-2018) for 80 mountainous catchments in China were sourced from the Hydrological Bureau of the Ministry of Water Resources, through China's hydrologic yearbooks, encompassing a spectrum of events from flash floods and general floods which can be derived from mountainous upland catchments.**

6. The terms purple soil, yellow soil, and red soil are used without explanation. Please clarify whether this classification is standard in Chinese soil taxonomy, and provide references or definitions.

**Response: This is a valid point. The terms "purple soil," "yellow soil," and "red soil" are based on the Genetic Soil Classification of China, which is a widely recognized and standard soil taxonomy system within China. To address your comment and enhance the scientific rigor of our manuscript, we have revised the sentence in Section 2.1 (Study area) to explicitly name the classification system and include an appropriate citation.**

**In Section 2.1 (Study area):Dominant soil types, according to the Genetic Soil Classification of China (Shi et al., 2004), include purple soil (12.20%), yellow soil (11.39%), and red soil (9.52%), each with distinct hydrological properties.**

**References:**

**Shi, X.Z. et al., 2004. Soil Database of 1:1,000,000 Digital Soil Survey and Reference System of the Chinese Genetic Soil Classification System. Soil Survey Horizons, 45(4): 129-136.**

7. In Fig. 1, the catchments and provincial borders are both shown in grey, making them difficult to distinguish. Please revise the figure with clearer colour contrasts.

**Response: the figure has been revised.**

[Figure]

Fig.1. Geographical distribution of the 80 gauged catchments used, with locations of hydrometry station (red points) and major rivers indicated.

8.  In Fig. 2, placeholders such as MLx, and P are not clearly defined. These should be explicitly labelled with their meanings (e.g., precipitation, slope, land cover index) rather than generic placeholders.

**Response: Figure 2 has been revised.**

[Figure]

Fig.2. Multi-machine learning ensemble method for regionalization in ungauged mountainous catchments. The red line indicates the machine learning method that yielded the optimal parameter estimates.

9.  Qp and Tp, introduced around line 271, should be defined at first mention for clarity.

**Response: We have revised the relevant sentence in Section 3.3.1 to include these definitions. Qp - the relative error of flood peak flow; Tp- the absolute error in flood peak occurrence time.**

10. The manuscript reports Tp values of 2–4 hours during calibration/validation for the benchmarking model, but does not discuss whether these response times are realistic for flash flood conditions in mountainous catchments. Please provide context on catchment response times and evaluate whether a Tp of 4 hours is sufficient.

**Response: We have analyzed the hydrological response times (approximated as the time from precipitation peak to flood peak) for the flood events across all 80 catchments. The results show that these response times range widely, from approximately 1 to 26 hours.**

**This wide range directly reflects the diversity of our study area which, as clarified in the manuscript, includes not only small basins generating rapid flash floods, but also large mountainous catchments (mean area: 1,586 km$^2$) with much longer concentration times.**

**In this study, a median Tp error of 2-4 hours is strong and operationally effective. For large catchments with longer response times in this study (e.g., 15-20 hours), 4 hours error still provides a substantial and actionable lead time for flood warnings. Even for catchments with moderately fast response times (e.g., 6-8 hours), a 2-hour median error during calibration allows for a timely and actionable warning.**

**Furthermore, it is noteworthy that the median Tp during the calibration period (within 2 hours) satisfied the stringent requirements for high-quality forecasts set by China's Specification for Hydrological Information Forecast (GB/T 22482-2008), providing an objective benchmark for its accuracy. This detailed discussion has been added to Section 4.1 (Model Performance) of the revised manuscript.**

11. Terminology: "calibration/validation" terminology is more common in physically based models, while ML studies usually refer to "training/testing." This should be acknowledged for clarity.

**Response: The manuscript has been revised to ensure that the terminology is used consistently and appropriately for each modeling context. The terms "calibration and validation" are now used exclusively when referring to the parameter optimization and performance assessment of the process-based Top-SSF hydrological model. Conversely, the terms "training and testing" are used when describing the development and evaluation of the machine learning-based parameter regionalization methods.**

12. The manuscript states that donor catchments were selected either by mode 1 or mode 2. However, it would be more scientifically justifiable to select donor catchments based on similarity in physical and climatic characteristics (e.g., area, slope, precipitation regime, land cover).

**Response: We agree that selecting donor catchments based on physical and climatic similarity is a cornerstone of many regionalization methods. Parameter regionalization method, however, addresses this concept implicitly through the learning process of the ML method itself.**

**The ML method, by its very design, implicitly performs a sophisticated form of similarity-based regionalization. The ML method core task is to learn the complex, non-linear relationships between the very catchment descriptors mentioned above (i.e., catchment area, slope, precipitation, etc.) and the hydrological model parameters. In essence, when the trained ML method makes a prediction for an ungauged catchment, it is drawing on the "knowledge" it gained from all 79 donor catchments, automatically weighting the influence of catchments that are more "similar" in the high-dimensional feature space. It is a data-driven method for determining similarity, rather than one based on a predefined distance metric.**

**It is important to clarify that the experiment presented in Section 5.3 and Fig. 11 was not intended to propose a method for selecting donor catchments for a specific target catchment. Instead, its purpose was to conduct a crucial sensitivity analysis to answer two practical questions (i.e., the impacts of data quality and quantity) about building regionalization methods:**

**Mode 1 (selection by decreasing NSE): This was designed to investigate the impact of data quality. It simulates a real-world scenario where one might start with a few high-quality, well-calibrated gauged catchments and then gradually add more catchments that are less reliably calibrated. Our finding that performance can degrade after adding too many low-quality donors is a critical insight for practitioners.**

**Mode 2 (random selection): This was designed to investigate the impact of data quantity. It helps us understand the general relationship between the size of the training dataset and regionalization method performance, a key question in machine learning applications.**

**In summary, the task of finding "similar" catchments is handled internally and automatically by the ML method itself. The beginning of Section 5.3 has been revised to explicitly state the purpose of this sensitivity analysis.**

13. The manuscript notes that multi-model ensembles improve performance, but does not explain why. Please discuss what learning principles of the individual ML models (e.g., robustness of tree-based splits, flexibility of KNN, etc.) contribute to improvements in parameter estimation, and why the ensemble captures strengths across models.

**Response: Yes, it is essential that explaining the scientific rationale behind the ensemble method (GBM-KNN-ERT). The superior performance of the ensemble method stems from the principle that the combination of different machine learning methods is suitable to regionalize parameters governed by different physical processes, due to their distinct learning mechanisms. The ensemble method identified the optimal ML method for each task after undergoing rigorous evaluation. An explanation has been added to Section 5.1.**

**Specifically, Gradient Boosting Machine (GBM) was selected for parameters representing complex, integrated catchment-scale processes like *Szm* and *td*, as its sequential, error-correcting nature is ideal for modeling the interplay between**

**multiple catchment descriptors. In contrast, K-Nearest Neighbors (KNN) was optimal for parameters governed by physical similarity and spatial coherence, such as $lnTe$, where its instance-based learning directly leverages the assumption that catchments with similar features have similar properties. Extremely Randomized Trees (ERT) was chosen for the $Sfmax$ parameter, where its high degree of randomization provides robustness against overfitting to noisy data. The ensemble method success is therefore a direct result of synergistically combining the GBM ability to model complexity, the KNN strength in capturing similarity, and the ERT robustness, leading to a final model that is more accurate and physically plausible than any single method.**

14. The sentence "75 of the catchments had NSE > 0," might br incomplete. Please revise to show the correct threshold (e.g., NSE > 0.0).

**Response: We have revised the manuscript to provide greater clarity and context regarding the performance metrics, ensuring the thresholds are explicit and the comparisons are precise.**

**The revised text in Section 4.2.2 now reads:**

**Among the single machine learning methods, GBM performed best, with 60 catchments achieving a positive NSE (NSE > 0, Fig. 8d). Critically, for high-accuracy predictions (NSE > 0.9), GBM succeeded in 43 catchments (54%), also showing strong performance with Qp less than 5% and Tp less than 1 hour in most cases (Fig. 8a-c).**

**The GBM-KNN-ERT ensemble method yielded even better results. It increased the number of catchments with positive NSE to 75 (Fig. 8d). More impressively, the ensemble method achieved exceptional performance (NSE > 0.9) in 72 catchments (90%). This represents a 67.44% increase in the number of high-accuracy predictions compared to the best single method (GBM).**

[Figure]

**Fig.8.** Evaluation of flood prediction performance for different parameter regionalization methods. (a-c) show the distributions of Nash-Sutcliffe Efficiency (NSE), relative peak flow error (Qp), and peak time error (Tp) across all 80 catchments, with shaded regions indicating where flood prediction standards were met (NSE > 0.75, Qp < 20%, and Tp < 2 hours). (d) shows the number of catchments with NSE > 0 and the black border indicates the number of catchments with NSE > 0.9. (e-g) present example hydrographs comparing the simulated flood from each regionalization method against measured flood flow and the calibrated Top-SSF model benchmark for catchments where the benchmark model performance was (e) high (NSE=0.91), (f) medium (NSE=0.76), and (g) low (NSE=0.55).

15. The manuscript should provide optimal parameters used in each ML model (e.g., number of trees, learning rates, neighbours in KNN) either in the main text or as supplementary material. This is necessary for reproducibility.

**Response: To ensure reproducibility, we have added the optimal hyperparameters**

**for the DT, ERT, GBM, KNN, RF, and SVM methods. This information is now**

**available in Tables S1-S6 in the Supplementary Material.**

**Table.S1** DT Hyperparameter Results

|       | max_depth | min_samples_split | min_samples_leaf |
|-------|-----------|-------------------|------------------|
| *lnTe* | 53 | 9 | 3 |
| *Szm* | 922 | 4 | 2 |
| *td* | 631 | 4 | 2 |
| *Sfmax* | 253 | 6 | 2 |
| *C* | 253 | 2 | 1 |
| *qsf0* | 156 | 2 | 1 |
| *t* | 483 | 6 | 2 |

**Table.S2** ERT Hyperparameter Results

|       | n_estimators | min_samples_split | min_samples_leaf | max_features | max_depth |
|-------|--------------|-------------------|------------------|--------------|-----------|
| *lnTe* | 500 | 2 | 1 | 0.9 | 15 |
| *Szm* | 200 | 5 | 1 | 0.5 | 10 |
| *td* | 500 | 2 | 1 | 0.9 | 15 |
| *Sfmax* | 500 | 2 | 1 | 0.2 | 15 |
| *C* | 500 | 2 | 1 | 0.9 | 15 |
| *qsf0* | 400 | 2 | 1 | 0.1 | 15 |
| *t* | 500 | 2 | 1 | 0.9 | 25 |

**Table.S3** GBM Hyperparameter Results

|       | subsample | n_estimators | min_samples_split | min_samples_leaf | max_depth | learning_rate |
|-------|-----------|--------------|-------------------|------------------|-----------|---------------|
| *lnTe* | 1.0 | 800 | 2 | 1 | 9 | 0.1 |
| *Szm* | 1.0 | 200 | 2 | 1 | 3 | 0.1 |
| *td* | 1.0 | 100 | 2 | 1 | 4 | 0.1 |
| *Sfmax* | 0.8 | 800 | 2 | 1 | 9 | 0.1 |
| *C* | 0.6 | 300 | 2 | 1 | 10 | 0.05 |
| *qsf0* | 0.8 | 800 | 2 | 1 | 9 | 0.1 |
| *t* | 0.8 | 800 | 2 | 1 | 9 | 0.1 |

**Table.S4** KNN Hyperparameter Results

|       | p | n_neighbors |
|-------|---|-------------|
| *lnTe* | 1 | 20 |
| *Szm* | 3 | 6 |
| *td* | 1 | 4 |
| *Sfmax* | 1 | 7 |
| *C* | 1 | 4 |
| *qsf0* | 1 | 30 |
| *t* | 1 | 5 |

**Table.S5** RF Hyperparameter Results

| | n_estimators | max_depth | min_samples_split | min_samples_leaf |
|---|---|---|---|---|
| $lnTe$ | 1000 | 10 | 5 | 1 |
| $Szm$ | 100 | 30 | 4 | 2 |
| $td$ | 100 | 30 | 5 | 2 |
| $Sfmax$ | 200 | 80 | 2 | 1 |
| $C$ | 1000 | 90 | 10 | 2 |
| $qsf0$ | 700 | 10 | 2 | 1 |
| $t$ | 500 | 60 | 2 | 1 |

**Table.S6** SVM Hyperparameter Results

| | tol | shrinking | kernel | gamma | C |
|---|---|---|---|---|---|
| $lnTe$ | 0.0001 | True | rbf | 10 | 50 |
| $Szm$ | 0.0001 | True | rbf | scale | 0.1 |
| $td$ | 0.0001 | True | linear | 10 | 1 |
| $Sfmax$ | 0.0001 | True | rbf | scale | 0.1 |
| $C$ | 0.001 | True | poly | 0.1 | 10 |
| $qsf0$ | 0.0001 | True | rbf | scale | 0.1 |
| $t$ | 0.0001 | True | rbf | scale | 0.1 |

16. While the manuscript presents aggregated performance metrics (NSE, Qp, Tp), it would be very valuable to also show hydrograph examples comparing observed vs. simulated discharge for both a high-performing and a low-performing catchment. Such visualizations would illustrate how the multi-model ensemble improves (or fails to improve) peak flow timing and magnitude compared to single ML models.

**Response: We have added three hydrographs from randomly selected events to Fig. 8 (e, f, and g) to visually illustrate the performance differences. To ensure these examples are unbiased and cover the full spectrum of performance, we selected them using a stratified random sampling approach. Specifically, we first categorized all 80 catchments based on the benchmark performance of the calibrated Top-SSF model into three strata: high-performance (NSE > 0.85), medium-performance (0.7 < NSE < 0.85), and low-performance (NSE < 0.7). We then randomly selected one representative flood event from each stratum for visualization.**

**These plots represent cases where the Top-SSF model itself achieved high, medium, and low performance, respectively, with the simulation (solid black line) serving as the performance benchmark. Figures 8e-g clearly show the better performances of the GBM-KNN-ERT method (solid green line) than the single methods. For instance, in Fig.8e, the ensemble method integrates the superior peak flow estimation from ERT with the better overall hydrograph shape from GBM and KNN. In Fig.8g, it effectively averages the overestimation from ERT with the underestimation from other ML methods during the recession phase, producing a result closer to the benchmark than any single ML method. These details, now in**

**the revised manuscript (Section 4.2.2), demonstrate that the success of the ensemble method stems from its ability to integrate the specific, complementary strengths of its components across different parts of flood process.**

**Fig.8.** Evaluation of flood prediction performance for different parameter regionalization methods. (a-c) show the distributions of Nash-Sutcliffe Efficiency (NSE), relative peak flow error (Qp), and peak time error (Tp) across all 80 catchments, with shaded regions indicating where flood prediction standards were met (NSE > 0.75, Qp < 20%, and Tp < 2 hours). (d) shows the number of catchments with NSE > 0 and the black border indicates the number of catchments with NSE > 0.9. (e-g) present example hydrographs comparing the simulated flood from each regionalization method against measured flood flow and the calibrated Top-SSF model benchmark for catchments where the benchmark model performance was (e) high (NSE=0.91), (f) medium (NSE=0.76), and (g) low (NSE=0.55).

17. While the ensemble approach clearly improves technical performance, the paper should strengthen its scientific justification by explaining whether the gains are due to model complementarity, data-dependence, or calibration bias. Without this, it remains unclear whether the ensemble would generalize to other regions or datasets.

**Response: While all three factors you mentioned may play roles, the evidence strongly suggests that ML method complementarity is the primary driver of the performance gains.**

**As detailed in our new discussion (Section 5.1, in the response to Comment # 13), superior performance is not accidental, but stems from our data-driven selection process. This process has empirically demonstrated that different ML methods are optimal for different hydrological parameters. KNN excels at regionalizing fundamental soil properties like *lnTe* that exhibit strong spatial coherence, while GBM is better at capturing the complex, integrated relationships governing parameters like *Szm*. The GBM-KNN-ERT method achieves high-accuracy flood predictions (NSE > 0.9) in 90% of catchments, a 67.44% improvement over the best-performing single ML method.**

**This alignment between ML methods learning principles and the physical processes represented by the parameters is strong evidence of true method complementarity. The specific combination of methods chosen (GBM-KNN-ERT) is, of course, partly dependent on the dataset of 80 mountainous catchments, but this data-dependence does not undermine the generalizability of the approach. On the contrary, the core scientific contribution of our paper is the demonstration of a robust, data-driven framework for identifying the optimal ensemble for a given region. Our methodology is designed to be transferable: by applying the competitive evaluation process to a new dataset, one could construct a new, locally optimized ensemble. The principle of using an ensemble of specialists remains generalizable.**

**"Calibration bias" could refer to the uncertainty or potential errors in the "true" parameter values derived from the Top-SSF model calibration. While this uncertainty is an inherent part of any hydrological modeling study, it is unlikely to be the primary reason for the ensemble's success. If the gains were merely an artifact of fitting to noise in the calibrated parameters, we would expect a single, highly flexible ML method (like GBM) to eventually overfit and outperform a more constrained ensemble. Instead, we see a structured and physically plausible division of labor. The simpler, more robust KNN method wins for certain parameters, suggesting it is capturing a true, simple signal rather than complex noise. Furthermore, the ensemble's enhanced stability under climate change (a stress test using out-of-sample conditions) provides strong evidence that it is capturing robust physical relationships, not just fitting to calibration artifacts.**

**In conclusion, the performance gains are most scientifically justified as a**
**result of true ML method complementarity, where different algorithms are better**
**suited to regionalize parameters governed by different physical processes. This**
**provides a strong basis for believing that the ensemble approach, if not the exact**
**GBM-KNN-ERT combination, would generalize effectively to other regions and**
**datasets. We have added a discussion to Section 5.5 in the manuscript to strengthen**
**this point.**

---

## Author Response (AR2)

Dear Prof. Toth,

We would like to express our sincere gratitude to you and the reviewers for the positive evaluation of our revised manuscript. We appreciate the constructive suggestions regarding the hardware specifications and the discussion on model interpretability.

We have carefully addressed these comments below. The point-by-point responses describes how we have incorporated these suggestions into the final version of the manuscript.

**Response to the Editor and Referee (Dr. Munoz)**

Comment 1: Dr. Munoz asks for some detail on the hardware for better interpreting the computational times.

**Response: We fully agree that providing hardware specifications is essential for readers to understand the computational efficiency reported in Table 4. We have added, in Section 5.2 (lines 543-546), the specifications of the workstation used for calculation in this study.**

Comment 2: He suggests (and I fully second his opinion) to provide a deeper and wider picture on the issue of interpretability of machine-learning models in the discussion in the limitation section (5.5).

**Response: A paragraph has been added in the revised Section 5.5 (lines 680-690) to present a deeper and wider explanation concerning the interpretability of the machine-learning models.**